# A Fe₃N/carbon composite electrocatalyst for effective polysulfides regulation in room-temperature Na-S batteries

Yuruo Qi[1,5], Qing-Jie Li[2,5], Yuanke Wu[1], Shu-juan Bao[1], Changming Li [1], Yuming Chen[2,3✉], Guoxiu Wang [4✉] & Maowen Xu [1✉]

The practical application of room-temperature Na-S batteries is hindered by the low sulfur utilization, inadequate rate capability and poor cycling performance. To circumvent these issues, here, we propose an electrocatalyst composite material comprising of N-doped nanocarbon and Fe₃N. The multilayered porous network of the carbon accommodates large amounts of sulfur, decreases the detrimental effect of volume expansion, and stabilizes the electrodes structure during cycling. Experimental and theoretical results testify the Fe₃N affinity to sodium polysulfides via Na-N and Fe-S bonds, leading to strong adsorption and fast dissociation of sodium polysulfides. With a sulfur content of 85 wt.%, the positive electrode tested at room-temperature in non-aqueous Na metal coin cell configuration delivers a reversible capacity of about 1165 mA h g$^{-1}$ at 167.5 mA g$^{-1}$, satisfactory rate capability and stable capacity of about 696 mA h g$^{-1}$ for 2800 cycles at 8375 mA g$^{-1}$.

[1] Key Laboratory of Luminescence Analysis and Molecular Sensing, Ministry of Education, Faculty of Materials and Energy, Southwest University, Chongqing 400715, PR China. [2] Department of Nuclear Science and Engineering, Massachusetts Institute of Technology, Cambridge, MA 02139, USA. [3] College of Environmental Science and Engineering, Fujian Normal University, Fuzhou 350007, PR China. [4] Center for Clean Energy Technology, University of Technology Sydney, Sydney, NSW 2007, Australia. [5]These authors contributed equally: Yuruo Qi, Qing-Jie Li. ✉email: yumingc126@126.com; Guoxiu.Wang@uts.edu.au; xumaowen@swu.edu.cn

Room-temperature sodium–sulfur (RT Na-S) batteries hold significant promise for large-scale application due to the abundance, nontoxicity, low cost and high theoretical capacity of both electrodes (Na anode and S cathode)[1–3]. Similar to Li–S batteries, Na–S batteries are also facing several challenges, such as low sulfur loading, unsatisfactory reversible capacity, inadequate rate capability, and rapid capacity fading[1–3]. These problems are typically triggered by the following intractable issues, which include the insulating nature of sulfur and its discharged products, the sluggish electrochemical reactivity of solid sulfur with sodium, the high solubility of polysulfides in the electrolyte[4–6] and the huge volume expansion during the discharge/charge process.

In attempting to address these problems, constructing hierarchical sulfur hosts is highly desirable. Nanocarbons[7–9] with excellent electronic conductivity and large surface areas are employed to balance the inferior electronic conductivity of sulfur and physically confine sodium polysulfides (NaPSs). However, the nonpolar feature of these carbon materials signifies insufficient mutual interaction to completely eliminate the shuttling of polysulfides. Subsequently, polar materials with intrinsic affinity to sulfur/polysulfides are introduced to anchor polysulfides chemically and thus enhance the cycling stability. Recently, electrocatalysts (including metals, metal oxides, metal sulfides, etc.) are proposed to accelerate the conversion kinetics of polysulfides and suppress the shuttling of polysulfides based on the strong catalytic activity[10–12]. Metal nitrides, featured with low cost, excellent chemical stability, good conductivity, and strong adhesion to polysulfides, have been widely investigated in Li–S batteries[13–15]; however, such metal nitrides have been rarely applied in RT Na–S batteries and their effectiveness and associated mechanisms in RT Na–S batteries remain unclear.

A further issue for most reported studies is the low sulfur content which could hinder the practical application. As Na–S batteries development is in its infancy, the attention to increase sulfur loading of Na–S cathodes is still insufficient. It has been acknowledged that 70% sulfur loading is critical to achieve high-energy-density Li–S batteries[16–19]. Being on a par with the energy density of Li–S batteries, a sulfur loading more than 70% in Na–S batteries is indispensable due to the larger atomic mass of Na. However, without proper cathode design, the massive load of sulfur will form a thick insulating layer, which eventually reduces the utilization rate and reaction kinetics. What's worse, the notorious shuttle effect of NaPSs will be exacerbated and result in worse cycling stability. Therefore, it is an urgent need to explore the rational design of hierarchical cathodes, which can synergistically inhibit the shuttle of NaPSs and enhance the electrochemical reactivity of sulfur with sodium under high-sulfur content.

In this research work, we propose a highly efficient $Fe_3N$ catalyst on N-doped multilayered carbon networks (NMCN) that enables strong adsorption and fast dissociation of NaPSs for advanced Na–S batteries. The NMCN featured with a hierarchical pore structure can accommodate massive sulfur, suppress volume expansion, and stabilize the cycling process. As far as we know, there has been no previous study employing nitrides in cathode hosts for Na–S batteries. The as-prepared $S@Fe_3N$-NMCN cathode can be directly used as a freestanding electrode without any other binders, conductive additives and current collectors, which significantly reduces the time and cost for assembling the battery. Both the pores within nanofibers and the interstice between layers can provide adequate space for sulfur loading. $Fe_3N$ shows prominent affinity to NaPSs via Na–N and Fe–S bonding, leading to efficient NaPSs dissociation. Benefiting from these structural and compositional advantages, a high-sulfur loading of 85wt.% ($2.6\ mg\ cm^{-2}$) is achieved. Importantly, the $S@Fe_3N$-NMCN cathode delivers satisfactory electrochemical performance in RT Na–S batteries, in terms of specific capacity ($1165.9\ mA\ h\ g^{-1}$ at $167.5\ mA\ g^{-1}$), rate capability ($658.4\ mA\ h\ g^{-1}$ at $16750\ mA\ g^{-1}$), and cycling stability (almost no capacity attenuation after 2800 cycles at $8375\ mA\ g^{-1}$). The robust hierarchical cathode design in this work demonstrates effectiveness in developing a stable cathode with high mass loading and stable kinetics, which can be generalized to other batteries systems. A simple graphical sketch of the Na–S cell proposed and the advantages of the $S@Fe_3N$-NMCN cathode are shown in Fig. 1a, which embodies the underlying reasons for the boosted performance.

## Results and discussion

**Characterization of $Fe_3N$-NMCN and $S@Fe_3N$-NMCN.** To obtain a promising reservoir for sulfur ($Fe_3N$-NMCN), commercialized bacterial celluloses (BC) infiltrated with $FeCl_3\cdot6H_2O$ were carbonized at 300 °C for 1 h and 800 °C for 2 h under a low-pressure $NH_3$ flow. The purpose of the treatment at the low temperature of 300 °C is to stabilize the structure by evaporating those volatile species in the BC precursor (such as CO, $CO_2$, methanol, and acetic acid)[20], before the final carbonization of BC at a high temperature of 800 °C. XRD and Raman results in Supplementary Fig. 1a–b confirm that the bare BC precursor can be precarbonized at 300 °C and the structure remains almost unchanged at the elevated temperature.

As listed by elemental analysis in Supplementary Table 1, the pristine BC consists of 46.86 wt.% C, 0.12 wt.% N, 46.37 wt.% O and 6.65 wt.% H. The small amount of "intrinsic" nitrogen atoms (0.12 wt.%) come from the residual nitrogen-containing compounds left by the culture media and secretions. When treated at 300 °C, most of the O/H volatilizes and a large amount of N (31.45 wt.%) is doped. As the temperature goes up, the amount of N/O/H reduces while that of C increases remarkably. Finally, the content of C, N, O and H in BC-800 are 89.77 wt.%, 6.45 wt.%, 2.56 wt.% and 1.22 wt.%, respectively.

XPS analyses (Supplementary Figure 1c and Supplementary Table 2) manifest that the amount of O=C species increases while those of O–C and O–C–O decrease dramatically when the temperature is raised. The final BC-800 is characterized by dominant O=C groups which have been reported to boost the affinity between the carbon anode and sodium ions[21,22], a favorable feature for performance enhancement. Meanwhile, pyridinic and pyrrolic N are two main N functional groups that can strongly bond to polysulfides and increase electronic conductivity[23–25]. FESEM images in Supplementary Figure 1d show that these composite fibers become thinner with the increasing temperature. In addition, the $FeCl_3\cdot6H_2O$ turns into $Fe_2O_3$ under the low-temperature treatment (around 300 °C) and then fully converts into $Fe_3N$ at 800 °C (Supplementary Fig. 2).

The as-obtained $Fe_3N$-NMCN composite retains the multilayered morphology of the pristine BC (Fig. 1b and Supplementary Fig. 3a–c) and maintains a self-supporting structure (inset of Supplementary Fig. 3a) with a thickness around 350 μm (Supplementary Fig. 3d). Each layer is interconnected by nanosized carbon fibers with a wide diameter distribution around 10–100 nm (Fig. 1c and Supplementary Fig. 3e–f). The TEM images in Fig. 1d and Supplementary Fig. 1g suggest that nanoparticles with a size of 3–10 nm are tightly implanted in these carbon nanofibers. The intertwined self-standing multilayer carbon networks can create a continuous pathway for electrons/ions (the electronic conductivity of carbonized bacterial cellulose aerogel can reach $20.6\ S\ m^{-1}$ [26]) and accommodate the volume change of active materials. Plentiful macropores generated by the interconnection of carbon layers (Fig. 1b and Supplementary Fig. 3h–i) can offer more voids to achieve high-sulfur loading.

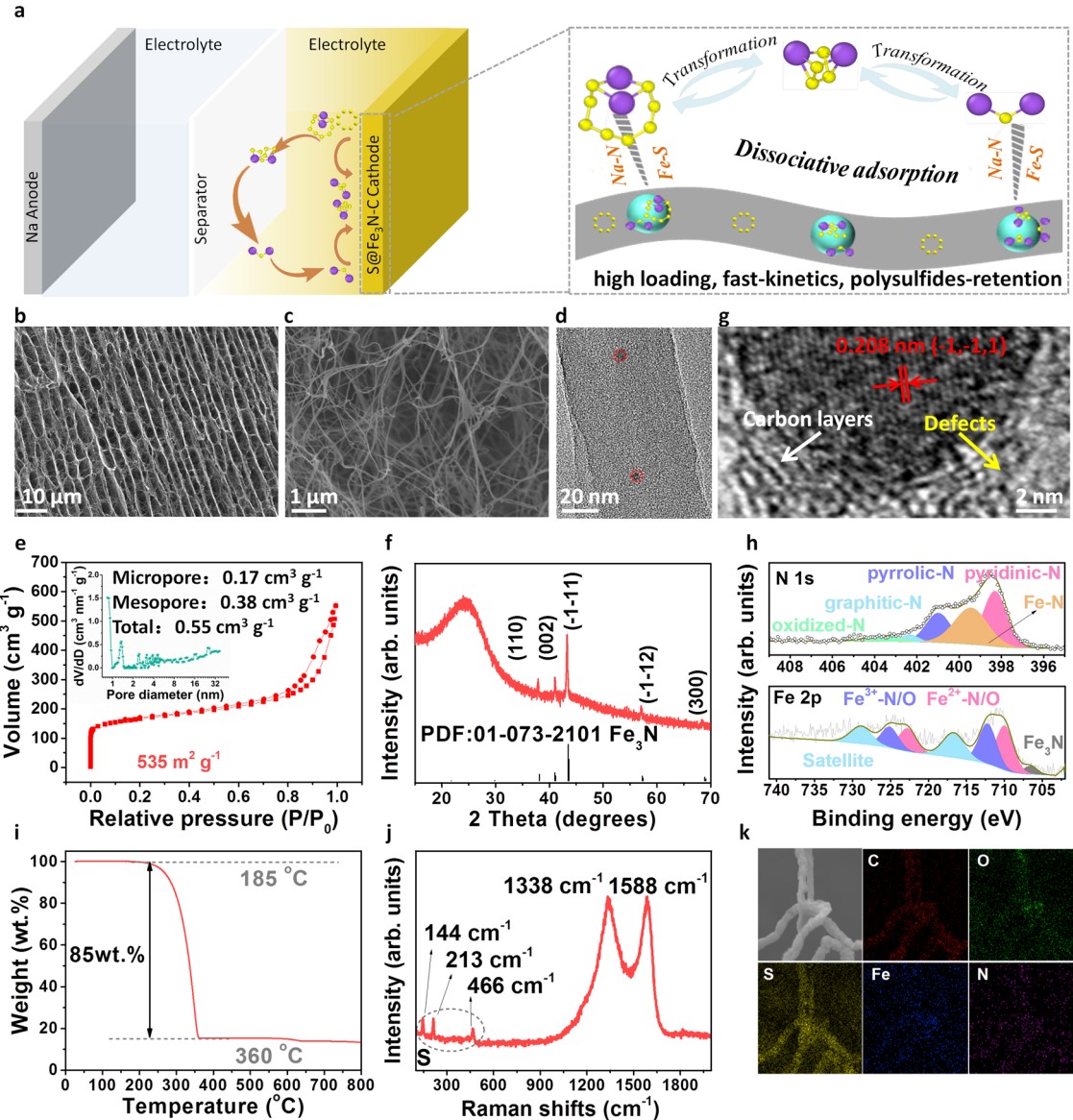

**Fig. 1 Characterization of Fe₃N-NMCN and S@Fe₃N-NMCN.** a Graphical sketch of the current Na–S cell and the function of the S@Fe₃N-NMCN cathode. **b–h** SEM (**b–c**), TEM (**d**), adsorption isotherm (**e**), pore-size distribution (inset of **e**), XRD (**f**), HRTEM (**g**), N 1s and Fe 2p spectra (**h**) of Fe₃N-NMCN. **i–k** TG curve (**i**), Raman spectrum (**j**), and EDS mapping images (**k**) of S@Fe₃N-NMCN.

Brunauer-Emmert-Teller (BET) analysis and corresponding pore-size distribution were used for a more detailed investigation of the porous structure, as shown in Fig. 1e and Supplementary Fig. 4. The BET surface area, total pore volume, micropores volume, and mesopores volume of the Fe₃N-NMCN composite, BC carbonized in Ar (abbreviated as BC-Ar) and BC carbonized in NH₃ (abbreviated as BC-NH₃) are compared in Supplementary Table 3. The BET-specific surface area of the Fe₃N-NMCN composite is as high as 535 m² g⁻¹, much higher than those of BC-NH₃ (425 m² g⁻¹) and BC-Ar (411 m² g⁻¹), providing prolific space for accommodating vast amount of sulfur. Moreover, the composite possesses a mixture of micropores (0.17 cm³ g⁻¹), mesopores (0.38 cm³ g⁻¹), and macropores (inset of Fig. 1e). It has been reported that the synergistic employment of hierarchical pores is an effective strategy to immobilize different types of S species and enable high-sulfur content[2,27,28]. Micropores are promising reservoirs for accommodating small S molecules and retaining NaPSs. However, the limited number of micropores always restricts the elevation of S content. Mesopores/macropores

can help enhance sulfur encapsulation and then realize a high S loading rate. Moreover, mesopores/macropores would greatly accelerate the electron/ion diffusion and offer enough space for the volumetric expansion of sulfur. Hence, the high specific surface area, in conjunction with the hierarchical pore structure, makes the composite a promising high-sulfur-content host.

The phase information of these nanoparticles was firstly investigated by X-ray diffraction (XRD) technique (Fig. 1f). Five characteristic peaks at 37.9°, 41.0°, 43.3°, 57.1°, and 68.7° are ascribed to the (110), (002), (−1−11), (−1−12), and (300) reflections of crystalline Fe₃N (PDF: 01-073-2101) with a hexagonal $P312$ space group[29]. The distinct lattice fringes with a d-spacing of 0.208 nm displayed by the high-resolution TEM (HRTEM) image (Fig. 1g) conforms to the (−1,−1,1) lattice plane of Fe₃N. The randomly oriented short carbon layers shown in Fig. 1g demonstrate the non-graphitized nature of the as-obtained carbon material. Besides, some defects can be observed mainly due to the N doping[30,31], as indicated by the yellow arrow. The disordered structure, together with abundant defects, contributes

favorably to the diffusion of sodium ions as well as the adsorption/catalytic reaction between $Fe_3N$ and NaPSs.

X-ray photo-electron spectroscopy (XPS) (Fig. 1h) is another indication of $Fe_3N$. In N 1s spectrum, a distinct Fe-N peak can be distinguished at 399.5 eV[32–34]. More importantly, Fe 2p spectrum exhibits a peak ascribed to $Fe_3N$[32–34] at the binding energy of 706.9 eV. Peaks at binding energies of 709.4/722.5 and 711.9/724.7 eV are originated from $Fe^{2+}$-N/O and $Fe^{3+}$-N/O, due to the surface oxidation of the sample. Peaks at binding energies of 715.8 and 728.9 eV are satellite peaks. The existence of $Fe_3N$ is also supported by the Raman spectrum in Supplementary Figure 5, in which peaks at 211, 280, and 383 $cm^{-1}$ are assigned to $Fe_3N$ according to the previous report[35]. The amorphous feature of carbon fibers is also revealed by the broad D/G bands at 1339/1591 $cm^{-1}$. Furthermore, the N 1s XPS spectrum verifies the existence of nitrogen-containing functional groups introduced during the $NH_3$ treatment process, including oxidized-N, graphitic-N, pyrrolic-N, pyridinic-N at binding energies of 403.6, 401.9, 400.7, 398.2 eV, respectively. It is believed that the doped N especially pyridinic and pyrrolic nitrogen can bond strongly to polysulfides and increase electronic conductivity[23–25]. The content of $Fe_3N$ in the $Fe_3N$-NMCN composite is 11 wt.%, according to TGA tests (Supplementary Fig. 6). In addition, based on the elemental analysis in Supplementary Table 4, the doped N in $Fe_3N$-NMCN is calculated to 9.36 wt.%, higher than those in BC-$NH_3$ (6.45 wt.%) and BC-Ar (0.34 wt.%). The XPS depth profiling results (Supplementary Fig. 7) after 0, 60, 120, 180 s of sputtering further demonstrate that both the N-doping and $Fe_3N$ are distributed uniformly through the whole matrix.

The above results suggest the successful embedding of $Fe_3N$ into N-doped multilayer carbon networks (NMCN), constructing a $Fe_3N$-NMCN composite as a high-loading host for Na–S batteries. First, the micropores/mesopores in nanofibers and the mesopores/macropores between layers can offer adequate space for sulfur loading, which guarantees the target of high-sulfur content. Second, the electronic conductivity $(0.1–1.0 \times 10^{-5} \text{ S cm}^{-1})$[36], high ionic diffusion ability $(1.7–3.6 \times 10^{-13} \text{ cm}^2 \text{ s}^{-1})$[37] and polar surface of $Fe_3N$ nanoparticles would endow the as-proposed cathode excellent reaction kinetics and facilitate the immobilization/transformation of polysulfides. Third, the nano-sized N-doped carbon fibers can offer interconnected conductive networks for the rapid migration of electrons and assist the adsorption/conversion of NaPSs.

Thus, we can load very high-sulfur content up to 85 wt.% (2.6 mg $cm^{-2}$) into the $Fe_3N$-NMCN composite (Fig. 1i) through a melting-diffusion method. The sulfur loading allows high specific energy for practical applications, outperforming other studies in which sulfur loadings are below the 50 wt.% range. Moreover, it is believed that further strategies to compress the porous S@$Fe_3N$-NMCN cathode can render a close packing of carbon fibers and therefore a thinner electrode, promising for high-energy density (Supplementary Fig. 8). The evident characteristic peaks of $S_8$ (PDF: 00-024-0733) were observed by XRD (Supplementary Fig. 9), which certifies the successful S deposition and adsorption in the $Fe_3N$-NMCN composite. The characteristic peaks of carbon (at 1338, 1588 $cm^{-1}$) and sulfur (at 144, 213, 466 $cm^{-1}$) were all detected by Raman (Fig. 1j) throughout the composite, verifying the complete and uniform penetration of sulfur through the material. In addition, a sharp decrease of specific surface area (32 $m^2 g^{-1}$) and pore volume (0.068 $cm^3 g^{-1}$) of the infiltrated material (Supplementary Fig. 10) suggests that a large proportion of pores are filled with sulfur. The residual pores will serve as an accommodation for the volume expansion during cycles.

FESEM and EDS mapping analyses of S@BC-Ar, S@BC-$NH_3$, and S@$Fe_3N$-NMCN were provided in Fig. 1k and Supplementary

Fig. 11. It is obvious that all three composites contain uniform distribution of sulfur without forming any bulk particles even under such a high S loading. The mapping images confirm again the uniform distribution of sulfur and also reveal that the self-standing structure is well preserved after S deposition. The above results demonstrate that a large amount of sulfur had been successfully embedded into the as-synthesized composites.

**Interaction between NaPSs and $Fe_3N$-NMCN**. The capability of the $Fe_3N$-NMCN composite to serve as an adsorbent for polysulfides was carefully probed (Fig. 2). Figure 2a compares the adsorption capability of 5 mg $Fe_3N$-NMCN composite, GO, Super P, BC-Ar, and BC-$NH_3$ in $Na_2S_6$-DOL/Diglyme solution (deep yellow color without any adsorbents). The color of the solution with Super P and BC-Ar remains almost unchanged, which is similar with most of the previous reports[27,28,38] that used non-polar carbon host materials. In contrast, a moderate adsorption ability to polysulfides is exhibited by BC-$NH_3$ (comparable to that of GO), as demonstrated by the slight decoloration feature of the solution including BC-$NH_3$, which would be resulted from the doping nitrogen (especially the pyridinic and pyrrolic N) in carbon fibers and its high surface area. Notably, the incorporation of $Fe_3N$ dots (the $Fe_3N$-NMCN composite) enables a significantly rapid color fading in only 10 min, indicating a strong adsorption ability for polysulfides. Density functional theory (DFT) calculations were carried out to elucidate the strong binding interplay between $Fe_3N$ and polysulfides. It is demonstrated that $Fe_3N$ can prompt strong adsorption and fast dissociation of sodium polysulfides. While the $Fe_3N$ dots play a dominant role in the adsorption of polysulfides, the contribution of doping N can not be ignored, as seen in the moderate color fading of the $Na_2S_6$-DOL/Diglyme solution (BC-$NH_3$). Together, the synergistic effect of $Fe_3N$ dots, doped N, and large surface area with abundant pores provides an effective strategy to address the issue of polysulfide shuttling.

UV–Vis spectra further support the above results, as shown in Fig. 2b. The original solution presents two broad bands at 424/612 nm that completely vanish after adsorbed by $Fe_3N$-NMCN for 24 h. These results prove the adsorption ability of the $Fe_3N$-NMCN composite to polysulfides, which can suppress the shuttling of polysulfides and ultimately enhance cycling stability. To verify whether the observed change in UV–Vis spectra is caused by adsorption or chemical reaction with precipitation of S-containing species, we analyzed the powders after 24 h of adsorption by XRD, SEM, and EDS. As shown in Supplementary Fig. 12a, the XRD patterns show no other peaks other than the (002) diffraction peak at ~26.0° (due to the amorphous carbon) and the peaks corresponding to $Fe_3N$ in $Fe_3N$-NMCN. This suggests that no new compounds form as a result of chemical reactions. Furthermore, SEM and EDS mapping images in Supplementary Fig. 12b–i demonstrate that there is no aggregation and all elements are uniformly dispersed in these substrates. The above results thus verify that the changes in UV–Vis spectra should be originated from the adsorption of $Na_2S_6$.

Furthermore, XPS spectra of the bare $Fe_3N$-NMCN powder and the $Fe_3N$-NMCN-$Na_2S_6$ composite were analyzed to investigate the underlying adsorption principles. An evident Na–N peak emerges at 1074.7 eV (Fig. 2c), besides the signal arisen from polysulfides (1071.7 eV). Concurrently, all peaks in the N 1s spectrum (Supplementary Fig. 13) move slightly to higher binding energies, although the shape of the spectrum remains basically unchanged. In addition, polar Fe-S bonds appear in the S 2p (159.3/160.5 eV, Fig. 2d) and Fe 2p spectrum (714.4/727.2 eV, Fig. 2e), which is helpful for immobilizing polysulfides. The peaks located at 164.1/165.4 eV in S 2p

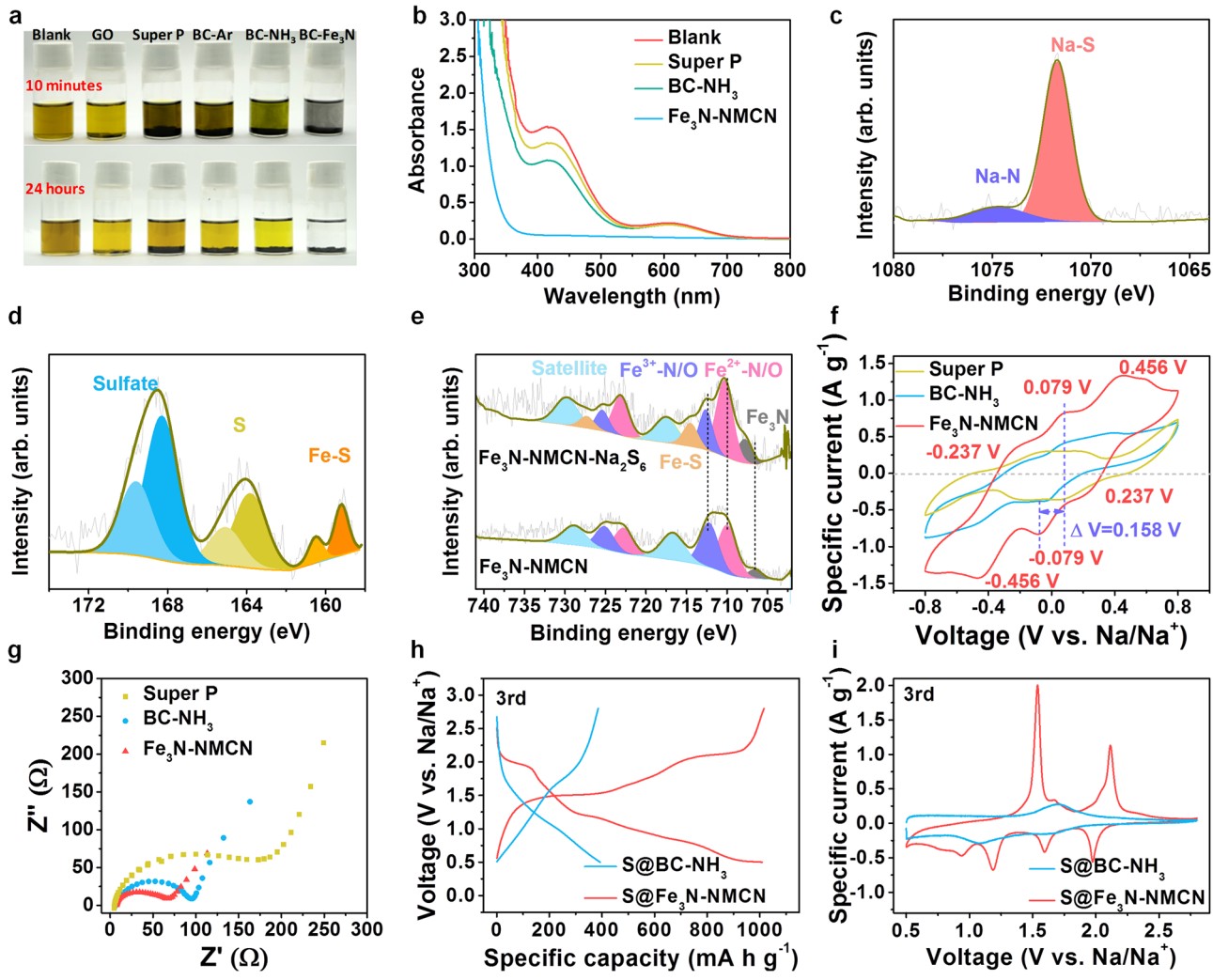

**Fig. 2 Interaction between NaPSs and Fe₃N-NMCN. a** Optical images of the blank Na₂S₆ solution and solutions with Fe₃N-NMCN, BC-NH₃, BC-Ar, GO, and Super P after 10 min and 24 h. **b** UV–Vis spectra of the blank solution and solutions with Fe₃N-NMCN, BC-NH₃, and Super P after 24 h. **c, d** Na 1s (**c**) and S 2p (**d**) XPS spectra after infiltrated with Na₂S₆. **e** Comparison of Fe 2p XPS spectra before and after Na₂S₆ adsorption. **f, g** CV curves (**f**) and EIS spectra (**g**) of symmetric cells with Fe₃N-NMCN, BC-NH₃, and Super P as cathodes. **h, i** Comparison of voltage-capacity curves at 167.5 mA g⁻¹ (**h**) and CV curves at 0.2 mV s⁻¹ (**i**) between S@Fe₃N-NMCN and S@BC-NH₃.

spectrum correspond to S-S bonds, while peaks at 168.4/169.7 eV are from the oxidized sulfates. Based on the above analyses, the Fe₃N-NMCN exhibits both sodiophilic (Na–N bonds) and sulfiphilic (Fe–S bonds) affinity with NaPSs, making it an attractive Na–S host.

To evaluate the catalytic effect of Fe₃N-NMCN on polysulfides transformation, cyclic voltammetry (CV) curves of symmetric cells with an electrolyte formed by Na₂S₆-DOL/Diglyme were gathered (Fig. 2f). The symmetric cells with both BC-NH₃ and Super P exhibit low response currents without any staged peaks. In Metal-S batteries, well-defined redox peaks with small voltage differences signify excellent catalytic activity in the kinetic conversion of polysulfides. However, staged redox peaks in symmetric cells are rarely observed in the Na–S battery literatures[10]. In contrast, the CV of Fe₃N-NMCN shows evident current response and three pairs of well-defined peaks (at 0.24/ −0.24, −0.08/0.08, and −0.46/0.46 V, respectively). As Na₂S₆ is the only active reactant, the peak located at −0.08 V is assigned to the reduction from Na₂S₆ to short-chain NaPSs while the peak at −0.46 V is originated from the conversion of short-chain NaPSs to Na₂S. Consequently, the peak at 0.24 V corresponds to the

reduction of S₈ to Na₂S₆. Notably, the polarization of the Fe₃N-NMCN electrode (0.16 V) is as small as those reported in the literature for Li-S batteries[39,40]. Overall, the high current response, well-defined peaks, and small polarization between redox peaks all refer to the strong catalytic ability of the Fe₃N-NMCN electrode, which effectively accelerates the electrochemical transformation of NaPSs.

The resistance of the Fe₃N-NMCN electrode is low (71 Ω for Fe₃N-NMCN vs. 95 Ω for BC-NH₃ and 185 Ω for Super P, Fig. 2g). This figure certifies that the electronic conductivity and interfacial affinity between Fe₃N-NMCN and polysulfides are effectively enhanced. This would facilitate the surface diffusion and subsequent conversion of polysulfides. The accelerated dynamics is also reflected by discharge/charge curves of the as-assembled Na–S cells (Fig. 2h), where the S@Fe₃N-NMCN cathode shows smaller polarizations. Correspondingly, more staged plateaus along with larger specific capacities are exhibited by the S@Fe₃N-NMCN cathode. Likewise, the CV curves of S@Fe₃N-NMCN in Fig. 2i present sharper peak profiles with much larger currents, suggesting the kinetics advantage of S@Fe₃N-NMCN for the conversion process.

**DFT calculations and AIMD simulations**. To gain further insight on the catalytic performance of $Fe_3N$ in Na–S batteries, a theoretical investigation based on DFT calculations[41–45] was conducted. The (−1−11) plane of $Fe_3N$ was representatively chosen for calculation according to our XRD and TEM observations. The adsorption energy ($E_{ad}$) was defined as $E_{ad} = E_{sub+molecule} − E_{sub} − E_{molecule}$, where $E_{sub+molecule}$, $E_{sub}$, and $E_{molecule}$ are the total potential energies of the substrate adsorbing a molecule, the substrate and the isolated molecule, respectively. Numerous geometries of $Na_2S_n$ ($n = 1, 2, 4, 6$ and $8$) on $Fe_3N(−1−11)$ were investigated and typical configurations were given in Fig. 3a. The calculated adsorption energies of various sodium polysulfides species on the (−1−11) surface of $Fe_3N$ were collected in Fig. 3b. The molecular adsorption energy of $Na_2S$ on $Fe_3N$ centers at −5 eV, manifesting that $Fe_3N$ can strongly anchor $Na_2S$ on its surface, as compared with other samples, such as $NiS_2$[12], metal[46,47], Mxene[48], MOFs[49], and carbon materials[12,46,47,49,50]. It is also unveiled that the strong polar-polar interaction between NaPSs and $Fe_3N$ results in an obvious dissociation of $Na_2S_n$ molecules. Moreover, $Fe_3N$ is prone to facilitate the dissociation of long-chain polysulfides.

Futhermore, ab initio molecular dynamics (AIMD) simulation was preformed to unfold the dissociation process of $Na_2S_8$ following the adsorption on $Fe_3N$. Figure 4a–c shows the structure, differential charge density, and potential energy evolution during the dissociative adsorption of $Na_2S_8$ on $Fe_3N$ (−1−11) as a function of AIMD simulation time. Our AIMD simulation suggests that the dissociative adsorption is dominated by the debonding of S atoms from the molecule and subsequent rebonding to $Fe_3N$ surface, in an incremental fashion. As shown in Fig. 4a–b, as the AIMD simulation proceeds, an increasing number of S atoms, 1S (1.1 ps) → 3S (1.74 ps) → 4S (3.86 ps) → 8S (9.53 ps), debond from the molecule and form new bonds with the $Fe_3N$ substrate. Correspondingly, the total potential energy of the system drops in a series of steps (Fig. 4c). During the dissociation process, charges are transferred to directional regions near an S atom and three nearest Fe atoms, forming new bonds between the S atom and the $Fe_3N$ substrate, in line with the XPS result in Fig. 2c–e. All S atoms from $Na_2S_8$ have adsorbed on the $Fe_3N$ surface in ~9.53 ps and then the $Na_2S_8$ molecule is completely dissociated with an energy drop of about 20 eV. These findings highlight that the $Fe_3N$ delivers a powerful interaction with NaPSs and can catalyze the dissociation of polysulfides.

**Electrochemical energy storage performances of the Na–S@$Fe_3N$-NMCN cell**. We expect the above S@$Fe_3N$-NMCN with well-designed structure and critical functional components to be a desirable positive electrode active material for RT Na–S batteries. The redox behavior was first examined by CV with the voltage range from 2.8 to 0.5 V at a scan rate of 0.2 mV s$^{-1}$ (Fig. 5a). In the first cathodic process, only a broad peak is observed at the voltage range of 1.5–0.8 V, while several well-defined oxidative peaks (at 2.13, 2.01, 1.87, and 1.62 V) emerge in the following charge process. The redox behavior in the following cycles is very different from that in the first cycle, which might be rooted in the decomposition of electrolytes to form solid electrolyte interphase (SEI) and different reaction process due to the electrochemically activation of S/polysulfides[51]. As a contrast, the BC-$NH_3$ was loaded with sulfur and employed as a cathode (S@BC-$NH_3$) for Na–S batteries (Supplementary Fig. 14). The S@BC-$NH_3$ cathode delivers different CV curves (Supplementary Fig. 15) with S@$Fe_3N$-NMCN. Two broad peaks (1.56, 1.05 V) are observed during the first cathodic process, while a sharp peak at 1.90 V and a broad peak at 1.73 V arise in the following charge process. Besides, the S@BC-$NH_3$ cathode displays smaller current response and larger polarization. The above results indicate that the S@$Fe_3N$-NMCN cathode enables a more kinetically efficient redox reaction process and low intrinsic resistance.

Figure 5b is galvanostatic discharge/charge profiles of the S@$Fe_3N$-NMCN cathode at 167.5 mA g$^{-1}$ with an electrolyte/sulfur (E/S) ratio of 18.2 μL mg$^{-1}$, where the transition behavior is highly consistent with the CV results. It delivers a discharge capacity of 1270.6 mA h g$^{-1}$ and a charge capacity of 1165.9 mA h g$^{-1}$, reaching an initial Coulombic efficiency (ICE) of 92%. Note that the employment of the DOL/DIGLYME electrolyte can enhance the electrochemical performance of Na–S batteries as reported by previous literatures[52,53] and compared with carbonate-based electrolyte (Supplementary Fig. 16), as ether-based electrolytes always function better with metal anodes and form robust SEI. However, it was also recognized by Cui[52] that a simple S/C cathode cannot retain good cycling stability in the ether-based electrolyte due to the shuttle of dissolved polysulfides, and by Hassoun[53] that ether-based electrolytes mitigate neither the polysulfide dissolution nor the polysulfide shuttle. Therefore, the elaborately designed S@$Fe_3N$-NMCN cathode contributes to the performance improvement.

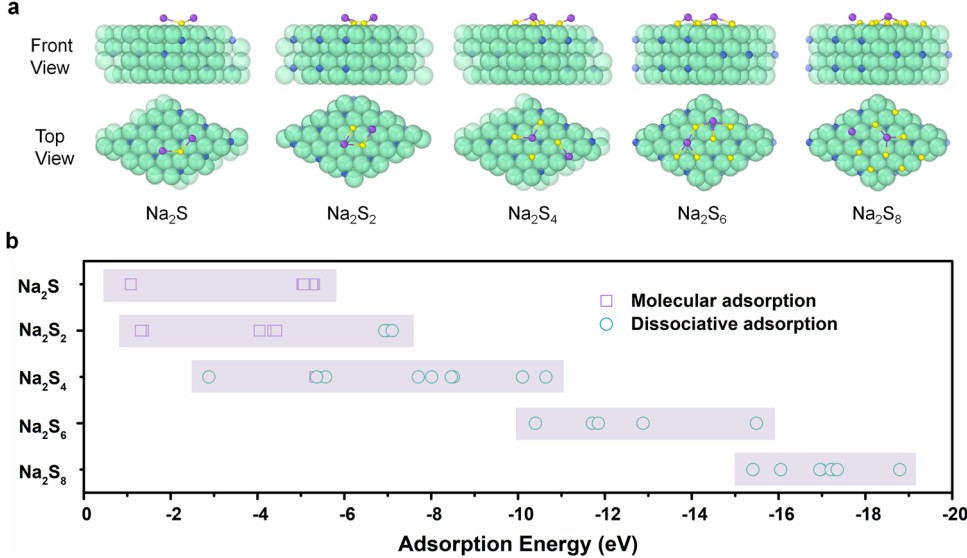

**Fig. 3 DFT calculations on the adsorption properties of $Fe_3N$. a** Typical adsorption configurations of $Na_2S_n$ ($n = 1, 2, 4, 6$ and $8$) on $Fe_3N$. **b** Calculated adsorption energies of various $Na_2S_n$ ($n = 1, 2, 4, 6$ and $8$) on $Fe_3N$.

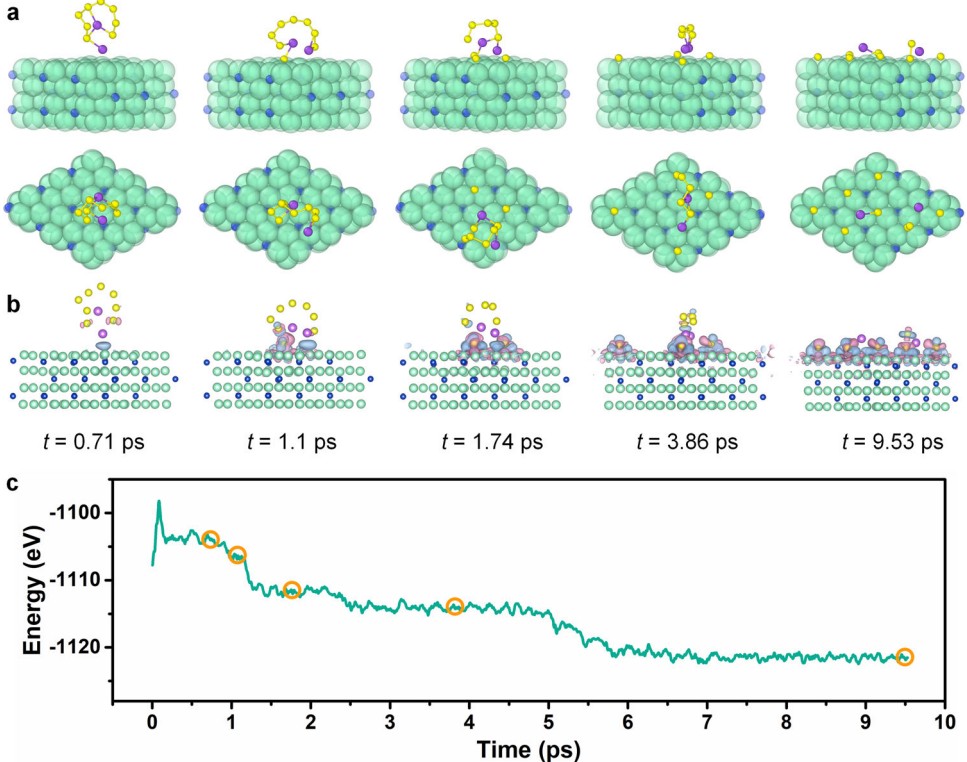

**Fig. 4 Theoretical analysis of the Na$_2$S$_8$ dissociative adsorption process.** Structure (**a**), differential charge density (**b**), and potential energy (**c**) evolution of Na$_2$S$_8$-Fe$_3$N as a function of AIMD simulation time. Blue/red isosurfaces in (**b**) represent electron accumulation/depletion regions, respectively. Isosurface scale level is set to 0.02 e/Å$^3$.

Compared with Na–S cells reported in literatures[7,10–12,27,48,54–61], the reversible capacity in this work is much closer to the theoretical capacity of sulfur. The capacity contribution from the Fe$_3$N-NMCN substrate is negligible (only 142 mA h g$^{-1}$ as illustrated in Supplementary Fig. 17a–b). Moreover, the cyclic curve of Fe$_3$N-NMCN in Supplementary Fig. 17c also demonstrates that the carbon network derived from BC can stabilize the cycle process. By contrast, the S@BC-NH$_3$ cathode shows smaller reversible capacity (413.0 mA h g$^{-1}$) and a low ICE (70%) without well-defined plateaus (Supplementary Fig. 18), demonstrating the crucial role of Fe$_3$N in improving S utilization rate. On the other hand, the S@BC-NH$_3$ cathode delivers no overcharge behavior despite of the modest capacity. However, when the S@BC-Ar composite (without N doping in the substrate) was employed as the cathode, a large overcharge capacity is observed (Supplementary Fig. 19), meaning that a severe shuttle of polysulfides has occurred[62]. This confirms that the doped N has a positive influence on bonding sulfur/polysulfides. Similar immobilization of polysulfides by the utilization of heteroatom doping has been widely reported for Li-S and Na–S batteries previously[23–25,28,54,63,64]. It should be emphasized that even under the identical preparation procedure (mass ratio between S and the substrate, temperature, etc.), smaller amounts of sulfur were immobilized in S@BC-NH$_3$ (74wt.%) and S@BC-Ar (62wt.%) compared with S@Fe$_3$N-NMCN (85wt.%). These results are well consistent with the BET data in Fig. 2a, where the specific surface area and pore volume of these substrates follow a decreasing order of Fe$_3$N-NMCN, BC-NH$_3$, and BC-Ar.

Good rate performance was also achieved by the well-designed S@Fe$_3$N-NMCN cathode, as displayed in Fig. 5c and Supplementary Fig. 20. Reversible specific capacities of 1238.6, 1073.7, 936.6, 866.2, 798.6, 706.9, 671.9, 664.2 and 658.4 mA h g$^{-1}$ are obtained at 167.5, 335, 837.5, 1675, 3350, 8375, 10050, 13400 and 16750 mA g$^{-1}$, separately. Notably, even at a rather high specific

current of 16750 mA g$^{-1}$, the reversible capacity retains 53%, among the best rate capability compared with previous reports[7,10–12,27,48,54–61,65]. What's more, the battery can almost restore to its initial capacity when going back to the low specific current of 167.5 mA g$^{-1}$. The voltage-capacity profiles (Supplementary Fig. 20) exhibit little hysteresis with increasing specific currents, further confirming the prominent reaction kinetics of as-assembled Na–S cells.

Electrochemical impedance spectroscopy measurements were carried out to investigate the resistance behavior of the sulfur-containing electrodes (Fig. 5d and Supplementary Fig. 21). The corresponding fitting results were listed in Supplementary Tables 5–6. Before cycle, the cell with S@Fe$_3$N-NMCN shows the lowest resistance (67.5 Ω) compared to S@BC-NH$_3$ (93.9 Ω) and S@BC-Ar (119.7 Ω). In addition, the cell with S@Fe$_3$N-NMCN displays the largest slope at the low frequency Warbug diffusion range, indicating the fastest diffusion of ions in the electrode. After cycled, the cell with S@Fe$_3$N-NMCN also exhibits the lowest SEI impedance (2.03 Ω) and charge transfer impedance (1.40 Ω). The enhanced performance of S@Fe$_3$N-NMCN compared to their counterparts can be attributed to the improved ionic diffusion and electronic conductivity of Fe$_3$N.

CV curves at various scan rates of 0.2, 0.5, 0.8, 1, 1.5, and 2 mV s$^{-1}$ were recorded (Supplementary Fig. 22) to investigate the reaction kinetics of the S@Fe$_3$N-NMCN cathode. The peak separation between each pair of redox peaks rises slightly with increasing scan rates (Supplementary Fig. 22a). Slopes calculated from plots of peak currents and square roots of scan rates (Supplementary Fig. 22b) are 393.2, 319.0, 324.5, −272.1, and −355.0 for A1, A2, A3, C1, and C2, respectively. In contrast, the S@BC-NH$_3$ cathode reveals inadequate performance at high specific currents (389.6, 363.0, 351.7, 329.8, 307.9, 295.7, 286.7,

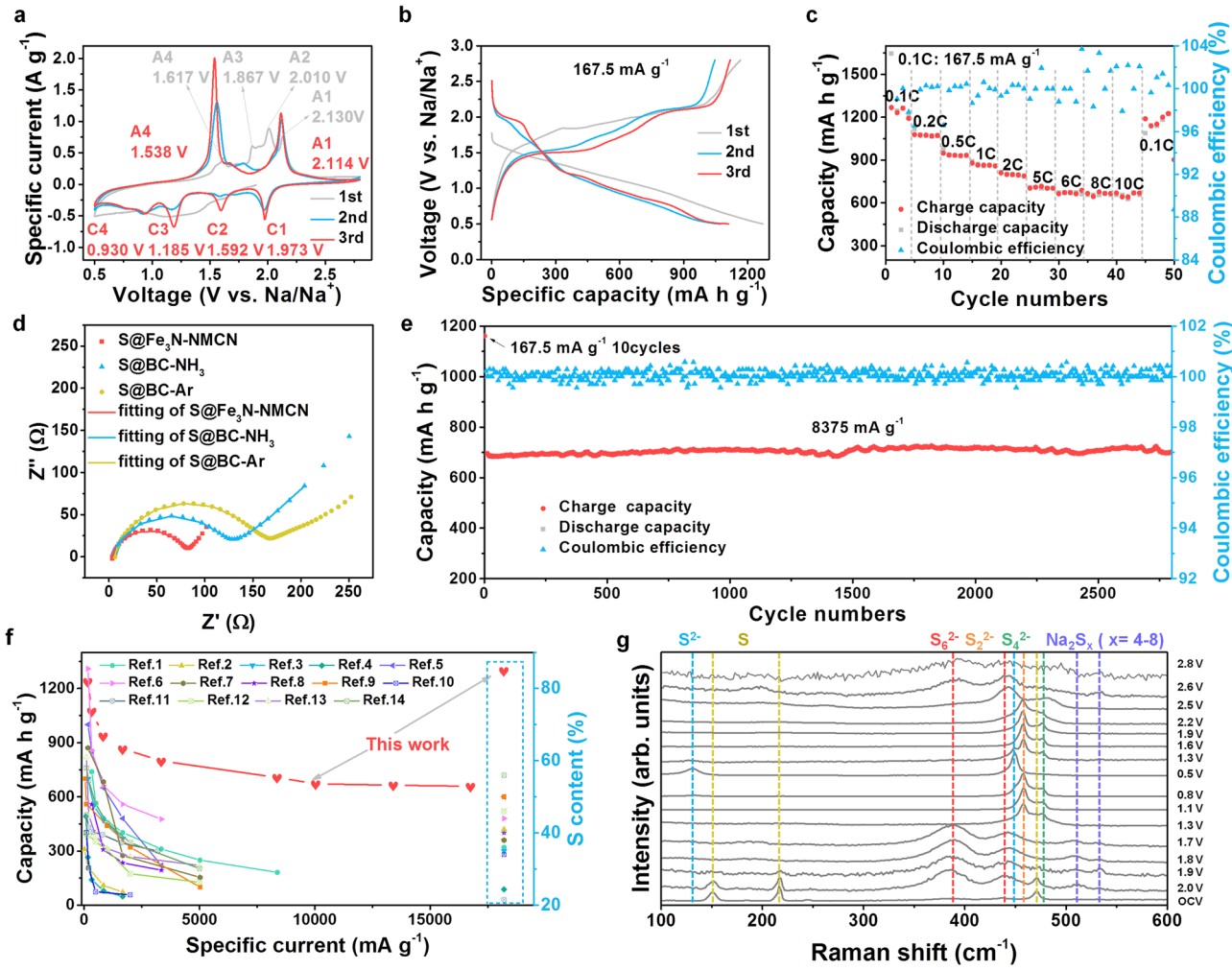

**Fig. 5 Electrochemical energy storage performances of the Na–S@Fe₃N-NMCN cell. a** First three CV curves at a scan rate of 0.2 mV s⁻¹. **b** First three voltage-capacity curves at a specific current of 167.5 mA g⁻¹. **c** Rate capability at specific currents of 167.5, 335, 837.5, 1675, 3350, 8375, 10050, 13400, 16750 mA g⁻¹. **d** Comparison of EIS spectra between BC-Ar, BC-NH₃, and Fe₃N-NMCN before the rate test. **e** Long-term cycling performance at a specific current of 8375 mA g⁻¹. **f** Comparison of electrochemical performance and sulfur content with previous literature. **g** In situ Raman spectra at a specific current of 167.5 mA g⁻¹.

281.7, and 294.1 mA h g⁻¹ at 167.5, 335, 837.5, 1675, 3350, 8375, 10050, 13400, and 16750 mA g⁻¹, respectively (Supplementary Fig. 23). Accordingly, the CV curves exhibit larger voltage hysteresis as the increase of scan rates (Supplementary Fig. 24a). Slopes from plots of peak currents and square roots of scan rates are much smaller (35.0 for A2 and −82.3 for C2 respectively, Supplementary Fig. 24b). As the diffusion coefficient is in direct proportion to the square of slope based on the Randles-Sevcik equation ($I = 2.69 \times 10^5 n^{1.5} A D^{0.5} v^{0.5} C$)[66,67], it is explicit that the S@Fe₃N-NMCN cathode possesses larger diffusion coefficients. The rate capability of the S@Fe₃N-NMCN electrode is promoted by the strong catalytic ability of Fe₃N as well as the long-range conductive networks formed by N-doped carbon nanofibers and Fe₃N quantum dots.

Another appealing property of the S@Fe₃N-NMCN cathode is the long-term cyclic stability. As illustrated in Fig. 5e and Supplementary Fig. 25, after cycled at 167.5 mA g⁻¹ for ten cycles, the reversible capacity at a rather high specific current of 8375 mA g⁻¹ can reach 696 mA h g⁻¹ and remain almost unchanged after 2800 cycles (698.7 mA h g⁻¹), far better than reported results[7,10–12,27,48,54–61,65]. Moreover, the battery can achieve a high and steady Coulombic efficiency of around 100%. The slight capacity variation can be due to the electrochemical

activation process (Supplementary Fig. 26 and Supplementary Table 7) and the ambient temperature change[68,69]. The cycling performance is achieved by the stable structure, good electrical conductivity and strong chemical interaction with NaPSs. From the comparison of the electrochemical performance with that of the current state-of-art Na–S batteries (Supplementary Table 8 and Fig. 5f)[7,10–12,27,48,54–61,65], the results obtained in this work are among the best performance reported, in terms of sulfur content, specific capacity, rate capability and cycling stability.

In situ Raman spectroscopy measurements (Fig. 5g and Supplementary Fig. 27) were performed to clarify the reaction process of the S@Fe₃N-NMCN cathode. Characteristic peaks of $S_8$[70] emerge at 151, 217, and 471 cm⁻¹ when the battery is in open-circuit voltage (OCV), and become weak after discharging to 2.0 V. Accompanied by the dissolution of $S_8$, a series of long-chain NaPSs appear at 390, 442, 508 and 534 cm⁻¹. Afterwards, at 1.3 V, two new peaks ascribed to Na₂S[50] and Na₂S₂[50] arise at ~478 and 458 cm⁻¹ respectively. Peaks of Na₂S[71] (at 131 and 449 cm⁻¹) emerge at 0.8 V and the intensity increases gradually until fully discharged to 0.5 V. In the following charge process, these peaks change oppositely from Na₂S to Na₂S₄, Na₂S₆, Na₂S$_x$ ($x = 4$–8); however, no signals of $S_8$ are discovered, in agreement with previous reports[72,73]. Different from previous reports that

peaks of $S_8$ exist throughout the whole discharge/charge process[10] and the deposited $Na_2S$ can still be detected in the full-charged electrode[71], a highly reversible reaction between $Na_2S_8$ and $Na_2S_2$ is presented in the as-proposed S@Fe₃N-NMCN cathode. In a whole, the discharge/charge process correlates to two distinct transitions from $Na_2S_8$ to $Na_2S_x$ (x = 4–8) in 2.8–1.3 V and then to $Na_2S$ in 1.3–0.5 V. As for the CV curves, the C1/C2 peaks may correspond to the conversion of long-chain NaPSs into short-chain, while the C3/C4 peaks may relate to the transformation of short-chain NaPSs to $Na_2S$.

We also carried out post-mortem analyses for the separators, the Na metal electrodes, and the as-employed electrodes disassembled from cells with S@BC-Ar, S@BC-NH₃, and S@Fe₃N-NMCN, respectively (Supplementary Figs. 28–33). Generally, the dissolution and shuttle of polysulfides lead to more yellowish separators. As can be seen, in contrast to the most yellowish separator from S@BC-Ar, the separator from S@BC-NH₃ shows less yellowish and no visible yellow color can be observed in the separator from S@Fe₃N-NMCN, suggesting increasingly inhibited dissolution and shuttle of polysulfides. For the cycled Na electrodes, dendrites are barely found in case of S@Fe₃N-NMCN when compared to their counterparts. In addition, a loose layer is formed on the cycled Na anode from BC-Ar, while the surface with S@Fe₃N-NMCN is smooth and clean. It has been reported[74] that a synergistic effect of lithium polysulfide and lithium nitrate in ether-based electrolyte can lead to a stable and uniform SEI layer on Li metal. However, it was also recognized that polysulfides alone cannot prevent the dendrite growth and minimize the electrolyte decomposition. In view of this, a better surface morphology of the Na anode for S@Fe₃N-NMCN compared with S@BC-Ar is resulted from the less shuttling of polysulfides to the Na anode as only polysulfides are presented in the electrolyte and no synergistic effect can be generated. The bonding ability to polysulfides achieved by Fe₃N, nitrogen doping, large surface area, and cathode porosity, can restrain the dissolved polysulfides in the cathode area. Thus, little polysulfides can enter into the anode area and deteriorate the Na anode. Supplementary Figs. 30–33 show that all three cycled electrodes can maintain the original morphology and all elements are uniformly distributed without aggregation; however, the S@BC-Ar electrode shows a smoother surface indicating the dissolution of some active sulfur materials into the electrolyte. The above results thus suggest that the dissolution and shuttle of polysulfides can be largely inhibited by Fe₃N-NMCN.

To further check the effect of electrolyte volume and the utilization of sulfur in Na–S batteries, we first studied the performance of the S@Fe₃N-NMCN electrode at different E/S ratios. As displayed in Supplementary Fig. 34a–b, in an E/S ratio range from 10.9:1 to 18.2:1, the capacities obtained in Na–S batteries are higher than most of the recently published literatures on Li-S batteries[13,14,69,75] and Na–S batteries[50]. In addition, the S@Fe₃N-NMCN electrode can still reach an acceptable capacity of 810.5 mA h g⁻¹ at a very low E/S ratio of 7.27 μL mg⁻¹. The capacity fading at low E/S ratios can be explained by the Na–S reaction mechanism that the electrochemical process is assisted by the dissolution of polysulfides and the fact that the increase of polysulfides concentration at low E/S ratios would result in the expansion of battery resistance.

Next, we studied the reaction process of the Na–S battery at a low E/S ratio of 7.27 μL mg⁻¹ by in situ XRD and ex situ XPS, as shown in Supplementary Fig. 35. Supplementary Fig. 35a–b demonstrate that the sulfur converts to $Na_2S_2$ when the discharge voltage drops to around 1.6 V. In the following charge process, the $Na_2S_2$ can be completely converted to $Na_2S_8$ again. Ex situ XPS spectra in Supplementary Fig. 35c–e lend further support to

the in situ XRD results. The original electrode shows a pair of peaks ascribed to sulfur at binding energies of 163.9/165.0 eV. When discharged to 0.5 V, two strong peaks of $S_2^{2-}/S^{2-}$ appear at 162.1 and 163.2 eV. The above results show that a series of highly reversible reactions between $Na_2S_8$ and $Na_2S_2$ can be achieved even at such a low E/S ratio of 7.27 μL mg⁻¹, demonstrating the potential of S@Fe₃N-NMCN to serve as a high-performance electrode for Na–S batteries.

It is worth mentioning that the proposed Fe₃N-NMCN substrate can also be applied to other multi-electron redox materials. For example, when 77% Se is incorporated (Se@Fe₃N-NMCN, Supplementary Fig. 36a), an average specific capacity of 671.4 mA h g⁻¹ is obtained at 70 mA g⁻¹ and can maintain 74% at 7000 mA g⁻¹ (496.4 mA h g⁻¹, Supplementary Fig. 36b–e). In addition, the S@Fe₃N-NMCN electrode is also applicable for Li-S batteries (Supplementary Fig. 37). In an electrolyte of 1 M LiTFSI in DOL/DME (1:1 by volume) with 0.1 M LiNO₃, the S@Fe₃N-NMCN electrode can deliver a high reversible capacity of 1345.7 mA h g⁻¹ with an ICE of 94.3% at 167.5 mA g⁻¹ (Supplementary Fig. 37a).

The good C-rate and long-cycle performance can be attributed to the synergistic effect of the N-doped carbon nanofibers and Fe₃N quantum dots. The carbon matrix generated from the carbonization of BC in NH₃ possesses several merits, such as large specific surface area, hierarchical pore, N doping, and good electronic conductivity. It can provide enough space to accommodate massive sulfur and adapt the volume change during cycling. The difference of electrostatic charges between the N-doped carbon matrix and polysulfides will also assist the entrapment of NaPSs to prevent shuttling. Furthermore, the electronic conductivity and ion diffusion of Fe₃N quantum dots improve the electrons/ions kinetics, while Fe/N atoms can form strong polar interaction with NaPSs and adsorb/catalyze NaPSs conversion. Besides, other alternative strategies such as the anode protection[74,76] and the electrolyte/separator modification[77,78] can be combined to further mitigate the detrimental effects of dissolved $Na_2S_x$, enabling a better performance and making the as-proposed electrode more practical.

In summary, Fe₃N quantum dots on N-doped multilayer carbon networks (Fe₃N-NMCN) were developed as an electro-catalyst targeting high-loading, fast-kinetics, and polysulfides-retention Na–S batteries. The N-doped multilayer carbon networks featured with large specific areas and hierarchical pores ensure the high-sulfur implementation. The N-doped substrate also provides the necessary physical adsorption due to the porous structure and extra chemical adsorption sites from the N doping. Meanwhile, the sulfur agglomeration and volume expansion are well suppressed by the multilayered carbon networks. In addition to the preceding advantages, the most substantial merit of this structure is the loaded Fe₃N quantum dots, which show strong affinity to polysulfides via Na–N and Fe-S bonds. The engineered multilayered carbon networks loaded with Fe₃N quantum dots efficiently inhibit NaPSs shuttling and promote NaPSs catalyzation. These features of the as-proposed cathode offer high reversible capacity (1165.9 mA h g⁻¹ at 167.5 mA g⁻¹), excellent rate capability (658.4 mA h g⁻¹ at 16750 mA g⁻¹), and stable cycling performance up to 2800 cycles, even at a practically relevant sulfur loading of 85wt.%. Most noteworthy, the cathode can be used as a self-standing electrode and the Fe₃N-NMCN host can also be extended to other multi-electron redox materials. This work not only clarifies the underlying mechanism for the improved performance, but also offers a strategy to design high-performance Na–S batteries. In addition to Fe₃N as identified in this work, we expect that many other nitrides can be explored as catalyst materials for Na–S batteries.

## Methods

**Preparation of Fe₃N-NMCN, BC-NH₃, and BC-Ar**. The commercial BC pellicles were first immersed into ethanol and deionized water for ultrasonic and then frozen in liquid nitrogen. In a typical synthesis of Fe₃N-NMCN, a piece of freeze-dried bacterial cellulose with a diameter of 19 mm was infiltrated by 50 μL ethanol solution of 25 mg mL⁻¹ FeCl₃·6H₂O. After drying at room temperature, these pieces were carbonized at 300 °C for 1 h and then 800 °C for 2 h under a low pressure of NH₃. The heating rate was 5 °C min⁻¹. A low temperature of 300 °C for 1 h was first employed to stabilize the BC structure and then finally carbonized at 800 °C for 2 h. After natural cooling to room temperature, the Fe₃N-NMCN film with a thickness of around 350 μm was obtained. BC-NH₃ and BC-Ar were obtained by the direct pyrolysis of BC under NH₃ and Ar atmosphere with the same temperature procedure as that of Fe₃N-NMCN. No purification step was employed after the pyrolysis in the synthesis of Fe₃N-NMCN, BC-NH₃, and BC-Ar. The weight of the Fe₃N-NMCN substrate was around 1 mg, while those of BC-Ar and BC-NH₃ were about 0.85 mg.

**Preparation of S@Fe₃N-NMCN, S@BC-NH₃, and S@BC-Ar**. 10 mg mL⁻¹ carbon disulfide solution of sublimed sulfur was dropwise added into Fe₃N-NMCN, BC-NH₃, and BC-Ar. The mass ratio of sulfur to the composite was kept at 10:1. After the volatilization of carbon disulfide in copper foil-wrapped crucibles at room temperature, it was heated to 155 °C with a temperature ramp of 2 °C min⁻¹ and kept at 155 °C for 12 h to ensure the well permeation of sulfur. After that, the temperature was elevated to 200 °C with a temperature ramp of 2 °C min⁻¹ and maintained at 200 °C for 30 min. Then, S@Fe₃N-NMCN, S@BC-NH₃ and S@BC-Ar were collected after cooling to room temperature. The weight of S@Fe₃N-NMCN was around 6.5 mg. The sulfur in the S@Fe₃N-NMCN electrode was around 5.5 mg. The diameter of S@Fe₃N-NMCN was around 16 mm. The mass loading of S in S@Fe₃N-NMCN was ~2.6 mg cm⁻². The electrolyte/sulfur (E/S) ratio for S@Fe₃N-NMCN was 18.2 μL mg⁻¹. The thickness of the S@Fe₃N-NMCN electrode was around 300 μm (Supplementary Fig. 38). To enhance the energy density of the electrode, an effective calendaring-infiltration process can be used to achieve a more compact electrode (Supplementary Fig. 8). The weight of S@BC-NH₃ and S@BC-Ar were around 3.2 and 2.3 mg. The sulfur in S@BC-NH₃ and S@BC-Ar were around 2.35 and 1.45 mg. The electrolyte/sulfur (E/S) ratio for S@BC-NH₃ and S@BC-Ar were 42.6 and 69.0 μL mg⁻¹.

**Adsorption experiment**. The Na₂S₆ solution (0.1 mol L⁻¹) was prepared by mixing sodium sulfide (Na₂S) and sulfur with a molar ratio of 1:5 in 1,3-dioxolane (DOL) and Diethylene glycol dimethyl ether (DIGLYME) (1: 1 Vol.%). Then, the solution was sealed with insert gas and stirred at room temperature for 20 h. 5 mg Fe₃N-NMCN, BC-NH₃, BC-Ar, GO and super P were added into the diluted Na₂S₆ solution, respectively, with the blank Na₂S₆ solution as a reference.

**Material characterization**. The morphologies of these samples were investigated by field-emission scanning microscope (FESEM, JSM-7800F, Japan) and transmission electron microscopy (TEM, JEM-2100, Japan). The EDS spectroscopy attached to FESEM was employed to record the elemental distribution. X-ray diffraction (XRD, Bruker, Advance D8A A25) was employed to investigate the phase information of the as-synthesized products. X-ray photoelectron spectroscopy (XPS) measurements were carried out on a Thermo Scientific ESCALAB 250Xi electron spectrometer. The sulfur contents in the as-prepared composites were determined by a Thermogravimetric analyzer (TGA, Q50, USA). The analyses for the specific surface area and the pore-size distribution were performed on micromeritics ASAP2020 equipped with V4.02 software under N₂ atmosphere. The Non-Local Density Functional Theory (NLDFT) model was used to calculate the d$V$/d$D$ vs. pore diameter curve. The pretreatment process was applied at 120 °C for 24 h. In addition, ex situ Raman was recorded by Lab-RAM HR Evolution Raman microscope with a 532 nm laser. The power of the 532 nm laser is 50 mW. The acquisition time per spectra is 24 s. The in situ Raman cell was purchased from EL-CELL company. The discharge/charge process of the in situ Raman was performed by a CV program on CHI760E. The corresponding CV curve was shown in Supplementary Fig. 27. Elemental analysis is conducted on Thermo Fisher (Flashsmart).

Ex situ characterizations of the Na₂S₆ infiltrated material after the adsorption test, the cycled eletrodes and the cycled Na metals were conducted according to the following steps. Firstly, these materials were filtrated, dried, and collected in the Ar glove box. Then, these materials were pasted on the platform and sealed with a plastic centrifuge tube. Thirdly, the platform was transferred into the chamber of the instrument and vacuumed quickly. Similarly, the XRD platform was sealed by a CAPTON tape.

**Electrochemical measurement**. To investigate the electrochemical energy storage performances, the S@Fe₃N-NMCN, S@BC-NH₃, and S@BC-Ar composites were directly used as cathodes to assemble coin-type (CR2032) cells in an argon-filled glove box (water and oxygen content lower than 5 ppm) with a calendering pressure of 50 kg cm⁻². Glass fibers (Whatman, GF1823-110) and sodium foils (Sinopharm Chemical Reagent Co., Ltd., 99.8%) were used as separators and the counter electrodes. The sodium foil was prepared manually during the battery assembly. The thickness of sodium foils is around 600 μm (Supplementary Fig. 39). 1 M NaPF₆ dissolved in 4-methyl-1,3-dioxy-cyclopentane/Diglyme (DOL/DIGLYME) (1:1 vol%) was employed as the electrolyte, whereas the water amount in the electrolyte is below 5 ppm. The amount of electrolyte added in each coin-type (CR2032) cell was 100 μL (an E/S ratio of 18.2 μL mg⁻¹). In addition, different amounts of electrolytes (80, 60, and 40 μL, different E/S ratios of 14.5, 10.9, and 7.27 μL mg⁻¹) were employed to further check the effect of electrolyte volume in Na–S cells. The in situ XRD was conducted under a low E/S ratio of 7.27 μL mg⁻¹ and thus the corresponding capacity of the in situ cell is 731.5 mA h g⁻¹, which is lower than that reported in the main text (1165.9 mA h g⁻¹ under an E/S ratio of 18.2 μL mg⁻¹). The in situ cell was purchased from Bruker (Beijing) Technology Co., LTD. A Be foil was used to seal the cell. The in situ XRD was conducted at a specific current of 167.5 mA g⁻¹ and an electrochemical window of 2.8–0.5 V on X-ray diffraction (XRD, Bruker, Advance D8A A25). Each XRD spectrum was collected within 12 min.

All the galvanostatic measurements were conducted on Land BT2000 battery test systems (Wuhan, China) in a voltage range of 2.8–0.5 V under room temperature (25 °C ± 5 °C) without any climatic/environmental chamber. CV test was carried out on CHI760E electrochemical workstation at scan rates of 0.2, 0.5, 0.8, 1, 1.5, and 2 mV s⁻¹. Zanner electrochemical workstation was used to collect the electrochemical impedance spectroscopy (EIS) measurements in a frequency range of 1 mHZ–1 MHZ (100 points per spectrum) with an amplitude of 5 mV. The correlation between the peak currents and the square roots of scan rates was determined by the Randles-Sevcik equation: $i_p = (2.69 \times 10^5)\ n^{1.5}\ A\ D^{0.5}\ C\ v^{0.5}$, where $i_p$ is the current at an electrochemical reaction, $n$ is the number of electrons participating in one mole of the reaction, $A$ is the electrode-electrolyte interface area (approximately to be the electrode area), $D$ is the apparent diffusion coefficient, $C$ is the mole concentration of Na⁺ in the electrode, and $v$ is the scan rate.

**First-principles calculation/simulation**. All ab initio calculations were carried out using the Vienna Ab initio simulation package (VASP). Electronic exchange-correlation functional was described by the generalized gradient approximation (GGA) in the form of Perdew-Burke-Ernzerhof (PBE). Core electrons were modeled with projector-augmented wave (PAW) pseudo-potentials (versions of Fe, N, S, and Na_pv as supplied in the VASP package) and a planewave energy cutoff of 520 eV was chosen. A k-point grid of 2 × 2 × 1 was used for the adsorption energy calculations and differential charge density analyses, while Γ point only sampling was performed for all molecular dynamics simulations (MD). Gaussian smearing with a width of 0.05 eV was used for the partial occupancies of states. All calculations are spin polarized. A slab configuration of Fe₃N in hexagonal crystal structure was used in all calculations. A vacuum space with thickness ≥10Å (depending on molecule positions) was inserted to model-free surface (close-packed plane in our case). The slab configuration contains 108 Fe atoms and 27 N atoms, with the bottom 2 layers fixed (in zero K adsorption energy calculations) or restricted to in-plane motion only (in 300 K ab initio MD simulations). Unrelaxed initial configurations for Na₂Sₓ ($x = 2, 4, 6, 8$) molecules were adopted from references. The initial adsorption configurations, with all molecule identities preserved by elastic constraints on bonds, were searched by the constrained minima hopping algorithm as implemented in the ASE package. Then molecules and surface atoms are fully relaxed to capture both molecular adsorption and dissociative adsorption processes. Ab initio MD simulations were carried out at 300 K under the $NVT$ (Nose-Hoover thermostat) ensemble. A time step of 1 fs was used for integrating the equation of motion. Visualization and differential charge density analyses were carried out using the Ovito package.

**Reporting summary**. Further information on research design is available in the Nature Research Reporting Summary linked to this article.

## Data availability

The data generated in this study are provided in the Supplementary information and main text.

## Code availability

This study does not use any custom code or mathematical algorithm, and code availability is not a mandatory requirement for this work.

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

## Acknowledgements
We appreciate support from National Natural Science Foundation of China (No. 22005251, No. 22179109), Chongqing Natural Science Foundation (cstc2020jcyj-bshX0047, cstc2020jcyj-zdxmX0010), Central Universities Fundamental Research Funds (XDJK2020C004, XDJK2019AA002).

## Author contributions
Y.Q. designed this work, synthesized the sample, analyzed the data, and wrote the paper; Q.-J.L. performed the theoretical calculation, analyzed the data, and revised the manuscript; Y.C., M.X., and G.W. provided sources, supervised the work, and revised the manuscript; Y.W., S.B., and C.L. participated the discussion.

## Competing interests
The authors declare no competing interests.
