## [Peer Review File · Nature Communications]

REVIEWER COMMENTS

Reviewer #1 (Remarks to the Author):

The approach of using metal nitride in hierarchical porous carbon is a well-known strategy to control the polysulfide dissolution for metal-sulfur batteries. However, the amount of sulfur loading of 85 % is not reported so far in the literature for RT NaS battery. This way it is interesting. The reported value of specific capacity and durability is higher when compared to previously reported NaS battery. The manuscript lacks many important details and should be modified (given below are the comments).

- What is the loading of Fe₃N in the composite structure S@Fe₃N-NMCN.?
- The loading of higher weight percent of sulfur can result in the agglomeration of sulfur. How uniformly sulfur is distributed in the composite?
- In the figure 5F, the cycling stability curve is observed to increase and then stabilises around 1500 cycles. Are there any side reactions that occur at the electrode? Surprisingly, the coulombic efficiency remains to be stable even at the increased cycle. Explain
- The initial coulombic efficiency for S@Fe₃N-NMCN is reported to be 92 % which is quite high for the metal-sulfur battery? Whether the electrolyte 4-methyl-1,3-dioxo-cyclopentane/Diglyme(DOL/DIGLYME) plays a role in enhancing the initial coulombic efficiency? The reported literatures on RT NaS battery uses carbonate and ether-based electrolytes, the electrochemical performance in carbonate-based electrolyte needs to be compared in order to validate the performance enhancement is solely due to the cathode design.
- Since the sulfurization process is carried out at the temperature of 155 C and are not heat treated to remove surface sulfur molecules, there is a possibility of more sulfur anchored onto the surface of the carbon matrix. The presence of surface sulfur will result in polysulfide shuttling. Evidence to be provided to confirm the sulfur is fully confined inside the carbon matrix rather than on the surface.
- The morphology of S@Fe₃N-NMCN, S@BC-NH₃ and S@BC-Ar needs to be compared in order to understand the distribution of sulfur in the composite.

Reviewer #2 (Remarks to the Author):

Reviewer's comments:

In the manuscript titled "Highly-efficient Fe₃N catalyst on N-doped hierarchical nanocarbons enables strong adsorption and fast dissociation of sodium polysulfides for advanced Na-S batteries", the authors reported Fe₃N catalyst on N-doped hierarchical nanocarbons for room temperature Na-S battery. The Fe₃N catalyst shows affinity towards sodium polysulfides via Na-N and Fe-S bonds, which result in a better Na-S cycling. Overall, the work is interesting and is relevant for the battery community. However, the manuscript lacks in few parts, which need to be further elaborated and discussed. Therefore, I cannot recommend for publication in Nature Communications in the current format. A major revision is recommended for reconsideration.

Below are my specific questions and comments.

1. The Fe₃N-NMCN was prepared in two steps, first at 300 °C for 1h and the second at 800 °C for 2h under NH₃ flow. I am curious why the synthesis proceeds in two steps? What is the mechanism for the carbonization of the bacterial cellulose and the growth of Fe₃N in NH₃ atmosphere? What is happening in the lower temperature and what at higher temperature? The reviewer suggests elaborating this part and to add a paragraph in the results and discussion to address these questions.
2. In the results and discussion only a vague and general sentence ("moreover, the composite possesses a mixture of micropores (D < 2 nm), mesopores (2 nm < D < 50 nm) and macropores (inset

of Figure 1f”) is presented about the pores. To have a better picture of the materials (Fe₃N-NMCN, BC-NH₃ and BC-Ar), please provide the total pore volume (TPV at P/P₀ = 0.99), pore volume for the micro and pore volume for meso. Which DFT model was used to calculate the dV/dD vs pore diameter? Does the surface area differ between the Fe₃N-NMCN, BC-NH₃ and BC-Ar? Was the BET measured with N₂ or other gases? Please provide the information in the experimental part.

3. The authors claim to have the N doping of the carbon materials, which is supported by EDS and XPS. However, there is no amount provided. How much of the N doping of the carbon is present in the Fe₃N-NMCN and BC-NH₃? Moreover, also the Fe₃N amount is missing. Please provide the amounts in wt.%. Are the N-doping and Fe₃N present on carbon surface or are embedded in the bulk of the material?

4. The authors claim that Fe₃N has “superior electronic conductivity and high ionic diffusion ability”. To what is these compared? How much are these values? Are these conclusions supported by EIS?

5. What was the Na₂S₆ concentration in the solution?

6. Did the authors perform a post-mortem analysis after prolong cycling? What was the separator color? How is look like the Na metal anode? The reviewer is curious what happened to the Na metal anode at a current density of 5C after prolong cycling and at 10C? Do Na dendrites form? How much is the Na anode preserved?

7. Coulombic efficiency and capacity vs cycle graphs should use reasonable y-axis scales. For instance, Coulombic efficiency should not be reported on a y-axis scale of 0-100% but rather 90-100%.

8. The authors claim that “It should be emphasized that even under the identical preparation procedure (mass ratio of S, temperature, etc.), smaller amounts of sulfur were immobilized in S@BC-NH₃ (74%) and S@BC-Ar (62%) compared with S@Fe₃N-NMCN (85%). These results are well consistent with the adsorption test in Figure 2a, where the adsorption ability of these substrates follows an order of Fe₃N-NMCN, BC-NH₃ and BC-Ar.” Is the melt infiltration of sulfur and the polysulfides adsorption two different physical phenomena? The melt infiltration is related to the filling of the total pore volume of the material. On the other hand, the polysulfides adsorption is the interplay the physical-chemical properties of the materials (different doping, surface area, porosity). As it was asked in the question 2, the total pore amount for all the materials should be provided to have a better comparison.

9. In the experimental part a lot of information’s are missing, such as:

I. Did the authors perform a purification step after the pyrolysis to remove some impurities? Please provide a sentence

II. How was the Na₂S₆ prepared?

III. What was the typical sulfur loading on the electrode? Please provide the areal mass (mg/cm²) loading of the sulfur active material

IV. What was the cathode thickness?

V. For the XPS measurements. Was any inert transfer for the materials which were in contact with the polysulfides?

VI. How the in-situ Raman cell look like? Please proved a schematics or reference. What was the power of the 512 nm laser and how much was the acquisition time per spectra? For the in-situ Raman, a galvanostatic graph for the in-situ discharge and charge should be provided.

VII. What was the sodium foil thickness? Was any pretreatment of the metal anode before the battery assembly?

VIII. To have a better comparison with the literature, please provide the normalized sulfur electrolyte volume ratio (the E/S in μL/mg S) not only the volume per cell.

IX. In the first-principles calculation/simulation experimental part please use the correct referencing. All the references from the first-principles calculation/simulate should be put in the reference section.

Minor issues:

1. There are some vague claims which are not supported by the data or references, such as “excellent electronic conductivity”.

2. Please tone down the word superior, it is used to frequently in the manuscript.

3. The authors showed an example of Se as other multi-electron redox materials. The reviewer is curious if Fe₃N-NMCN can be efficiently used also in Li-S and Mg-S batteries?

Reviewer #3 (Remarks to the Author):

This work reports an N-doped self-standing electrode using a Fe₃N catalyst for room-temperature (RT) Na-S batteries. The authors provided various experimental data on a high-performance Na-S cell benefiting from the proposed electrode composition, along with adsorption tests and simulations to investigate the catalytic effect of Fe₃N. The high capacity, cycling stability, and rate capability shown in this report might be of interest for the scientific community working in the field of RT Na-S batteries. However, the results provided by the authors do not clearly elucidate the actual role of Fe₃N on the electrochemical behavior of Na-S batteries, and the conclusions appear to be not properly supported by evidence. Therefore, we suggest resubmission to a more specialized journal after addressing the major issues listed below.

- 1) The authors reported a sulfur loading on the cathode of 2.6 mg/cm², which may be considered rather promising, although further improvements are certainly needed to enable practical applications (Joule 4, 285–291 (2020)). On the other hand, other crucial electrode metrics are missing, such as thickness and weight of the Fe₃N-NMCN support, as well as the electrolyte/sulfur (E/S) ratio. These parameters are needed for a proper evaluation of the cell performance, in particular considering the non-conventional, self-standing cathode configuration, and should be compared to those of the benchmark electrodes shown in Figs. S9, S11, S12. A lack of microstructural and cell metric data on both the proposed and the benchmark electrodes and does not allow to separate the effects of cathode morphology and Fe₃N catalyst.
- 2) The authors should further investigate the reasons behind the observed change in UV/Vis spectra of the catholyte solutions when in contact with the several electrode powders (Fig. 2a–b). XRD and SEM-EDS analyses of these powders should be performed to verify whether that change is due to adsorption or chemical reaction with precipitation of S-containing species. Moreover, XRD and SEM-EDS data of various electrodes after cycling in the cell should be provided to demonstrate the reversibility of the electrochemical reaction.
- 3) The change in voltage profiles during the long-term cycling test should be shown (Fig. 5f). Does the voltage curve vary after 2800 cycles?
- 4) Most of the XPS data have an unacceptably low signal-to-noise ratio, which significantly affects the reliability of the related analyses (for instance Figs. 1h and 2d–e). Similarly, the Raman spectrum of Fig. S3 shows very weak Fe₃N signals. These analyses should be repeated.
- 5) The F₃N ratio in the composite should be indicated in the manuscript.
- 6) This paper propose the use of the Fe₃N catalyst to achieve long-term cycling by mitigating the sulfur dissolution, although it does not provide sufficient experimental proof of the actual polysulfides (PSs) retention in the cathode during cycling. In this regard, it might be worth remarking that various recent papers have demonstrated the full dissolution of the Li-PSs in the electrolyte, for instance by operando X-ray microscopy and diffraction (Energy Environ. Sci. 11, 202–210 (2018) and Adv. Energy Mater. 5, 1500165 (2015)), as well as in situ EPR and NMR (J. Electrochem. Soc. 162, A474–A478 (2015) and Nano Lett. 15, 3309–3316 (2015)), thereby partially addressing the conversion mechanism models proposed in the first stages of the alkali metal-S battery research.

Reviewer #1

The approach of using metal nitride in hierarchical porous carbon is a well-known strategy to control the polysulfide dissolution for metal-sulfur batteries. However, the amount of sulfur loading of 85 % is not reported so far in the literature for RT NaS battery. This way it is interesting. The reported value of specific capacity and durability is higher when compared to previously reported NaS battery. The manuscript lacks many important details and should be modified (given below are the comments).

Response: We greatly appreciate your highly constructive comments and giving us the opportunity to improve our manuscript.

Comment 1: What is the loading of Fe_3N in the composite structure $\text{S}@\text{Fe}_3\text{N}\text{-NMCN}$?

Response: Thanks for the valuable question. We have conducted TGA tests to detect the loading of Fe_3N in the $\text{Fe}_3\text{N}\text{-NMCN}$ composite.

To clarify the transformation process of Fe_3N , TGA analyses of the bare Fe_3N powder and $\text{Fe}_3\text{N}\text{-NMCN}$ in O_2 atmosphere were conducted (Figure S8a-b). Moreover, the phase information of the final products after TGA tests was also detected by XRD (Figure S8c-d). The bare Fe_3N powder was obtained through the heat treatment of $\text{FeCl}_3 \cdot 6\text{H}_2\text{O}$ at 300 °C for 1 h and then 800 °C for 2 h.

As displayed in Figure S8c-d, both products obtained after TGA tests are Fe_2O_3 (159.6 g mol⁻¹). When we assume that there is 1 mol Fe_3N (181.4 g mol⁻¹), there should be 1.5 mol Fe_2O_3 after the TGA test since the mole number of Fe is constant. During this process, the weight increment is 132.0wt.% (=159.6*1.5/181.4). This result is highly consistent with the TGA curve of the bare Fe_3N powder in Figure S8a, where the weight increase between room temperature and 800 °C is about 134wt.% and the main weight variation is in the range of 350 and 800 °C.

Therefore, it can be deduced that the weight loss (89wt.%) of $\text{Fe}_3\text{N}\text{-NMCN}$ before 353 °C (Figure S8b) is exclusively originated from the consumption of carbon matrix in $\text{Fe}_3\text{N}\text{-NMCN}$ and is not related to Fe_3N , while the weight increment (132wt.%=14.5wt./11wt.%) between 353 and 800 °C is only from the oxidation of Fe_3N to Fe_2O_3 . From above analyses, it has been determined that the weight percentage of Fe_3N in $\text{Fe}_3\text{N}\text{-NMCN}$ is 11wt%.

Updates to the revised manuscript: We have added these results in the revised manuscript on Page 11, highlighted in yellow.

Figure S8. Calculation of Fe₃N content in the Fe₃N-NMCN composite.

(a-b) TGA tests of the bare Fe₃N powder (a) and the Fe₃N-NMCN composite (b).

(c-d) XRD patterns of the final products after TGA tests from the bare Fe₃N powder (c) and the Fe₃N-NMCN composite (d).

Comment 2: The loading of higher weight percent of sulfur can result in the agglomeration of sulfur. How uniformly sulfur is distributed in the composite?

Response: We appreciate the reviewer for the instructive comment. FESEM and EDS mapping analyses of S@BC-Ar, S@BC-NH₃ and S@Fe₃N-NMCN were conducted and the results are provided in Figure 1k, Figure S12. It is obvious that three composites contain uniform distribution of sulfur without any bulk particles formed even under such a high S loading. The mapping images further confirm the uniform distribution of sulfur and reveal that the self-standing structure has been well preserved after S deposition.

Figure S12. FESEM and corresponding EDS mapping images of S@BC-Ar, S@BC-NH₃ and S@Fe₃N-NMCN.

Updates to the revised manuscript: We have added these results in the revised manuscript on Page 12, highlighted in yellow.

Comment 3: In the Figure 5f, the cycling stability curve is observed to increase and then stabilize around 1500 cycles. Are there any side reactions that occur at the electrode? Surprisingly, the coulombic efficiency remains to be stable even at the increased cycle. Explain

Response: Thanks for this valuable question. The capacity increase during the whole long-term cycling process is -2%~4% (the initial capacity is 696 mA h g⁻¹, the largest and smallest capacities are 726.4 and 682.7 mA h g⁻¹, respectively). There may be several reasons. It is well accepted that the capacity increase in the first few cycles could stem from the activation of the electrode, as widely observed in previous reports^{21,23}. Furthermore, to verify the above speculation, the electrochemical impedance spectra during first 55 cycles at a current density of 83.75 mA g⁻¹ (ten times smaller than that in Figure 5f) were recorded, as shown in Figure S27. The fresh battery has a resistance of 71.25 Ω. The resistance declines dramatically to 6.86 Ω after 5 cycles (around 200 hours) and 0.89 Ω after 15 cycles (around 600 hours).

Moreover, the value of resistance maintains unchanged in the following cycles. The evolution of EIS indicates that the proposed electrode can reach stable state in about 600 hours. This coincides well with the Figure 5f, where a slight capacity increase is observed in initial 1500 cycles (with a ten times larger current density of 8375 mA g^{-1} , but the same cycling time of 600 hours).

Besides, the ambient temperature change would also result in the fluctuation of capacity.

Figure S27. EIS of batteries with S@Fe₃N-NMCN after different cycles at a current density of 83.75 mA g^{-1} .

Table S8. EIS fitting results of batteries with S@Fe₃N-NMCN after different cycles.

Sample	R_s (Ω)	R_1 (Ω)
pristine	3.95	71.25
After 5 cycles	2.70	6.86
After 15 cycles	3.06	0.89
After 25 cycles	2.69	0.75
After 35 cycles	3.02	0.71
After 45 cycles	2.97	0.81
After 55 cycles	2.42	1.00

Updates to the revised manuscript: We have added these results in the revised manuscript on Page 26, highlighted in yellow.

Comment 4: The initial coulombic efficiency for S@Fe₃N-NMCN is reported to be 92 % which is quite high for the metal-sulfur battery? Whether the electrolyte 4-methyl-1,3-dioxo-cyclopentane/Diglyme(DOL/DIGLYME) plays a role in enhancing the initial coulombic efficiency? The reported literatures on RT NaS battery uses carbonate and ether-based electrolytes, the electrochemical performance in carbonate-based electrolyte needs to be compared in order to validate the performance enhancement is solely due to the cathode design.

Response: We thank the reviewer for the constructive comment. Indeed, the performance enhancement and the high Coulombic efficiency are supported by the employment of the DOL/DIGLYME electrolyte. As exhibited in Figure S17a, a lower ICE of 73.4% is obtained in carbonate-based electrolyte (1M NaClO₄ in EC:DEC, 1:1 by volume). Moreover, the capacity decays rapidly in about 100 cycles (Figure S17b). It is still similar in Li-S batteries (Figure S38), where the S@Fe₃N-NMCN electrode can hardly work in the carbonate electrolyte (1M LiPF₆ in EC:DEC:EMC, 1:1:1 by volume) while it can deliver a high charge capacity of 1345.7 mA h g⁻¹ with an impressive ICE of 94.3% in an electrolyte of 1M LiTFSI in DOL/DME (1:1 by volume) with 0.1 M LiNO₃.

Figure S17. Discharge/charge curves (a) and cycling performance (b) of S@Fe₃N-NMCN with 1M NaClO₄ in EC/DEC (1:1 by volume) electrolyte in Na-S batteries.

Figure S38. Discharge/charge curves of S@Fe₃N-NMCN with 1M LiTFSI in DOL/DME (1:1 by volume) containing 0.1M LiNO₃ additive (a) and 1M LiPF₆ in EC/DEC/EMC (1:1:1 by volume) electrolyte (b) in Li-S batteries.

In addition, it should be noted that this phenomenon is not uncommon in metal-ion batteries and metal-sulfur batteries. For metal-ion batteries, ether-based electrolyte is found to function better with anode and form robust solid electrolyte interface (SEI), especially for alloying/conversion and carbon anodes. In metal-sulfur batteries, the improvement of electrochemical performance in ether-based electrolyte is also reported.

For example, Cui's group²⁴ reported that metallic anodes can work very well in a simple liquid electrolyte, sodium hexafluorophosphate in glymes (mono-, di-, and tetraglyme). It can enable highly reversible and nondendritic plating-stripping of sodium metal anodes at room temperature with high average Coulombic efficiencies of 99.9%. The long-term reversibility was found to arise from the formation of a uniform, inorganic solid electrolyte interphase that is made of sodium oxide and sodium fluoride, which is highly impermeable to electrolyte solvent and conducive to nondendritic growth. As a proof of concept, they also demonstrated a room temperature sodium-sulfur battery using 1 M NaPF₆ in tetraglyme with a simple S/C cathode as the working electrode. The simple S/C cathode exhibited a specific capacity of 776 mA h g⁻¹ and good cycling stability in 20 cycles. However, an obvious shuttle effect from dissolution of intermediate long-chain polysulfide species into the electrolyte was observed. Through the improvement of S cathode reported in our manuscript (a unique 3D carbon matrix embedded with Fe₃N catalyst as an effective host to trap the polysulfide species and minimize their dissolution into the electrolyte), the overall performance has been boosted in terms of Coulombic efficiency, specific capacity and cycling performance. The above result shows that both the glyme-based electrolyte and the as-obtained S@Fe₃N-NMCN electrode in this manuscript contribute a lot to the outstanding performance.

Figure R1. Cyclic voltammograms at 0.1 mV s⁻¹ (a), typical charge–discharge voltage profile (b) and cycling performance (c) of a room-temperature Na–S battery cycled using 1 M NaPF₆ in tetraglyme with a simple S/C cathode as the working electrode and Na metal as the counter and reference electrodes. [*ACS Cent. Sci.* **1**, 449-455,(2015).]

Hassoun²⁵ reported a capacity of 500 mA h g⁻¹ with a TREGDME-NaCF₃SO₃ electrolyte. However, the second and third cycles revealed a remarkable decrease of the delivered capacity accompanied by a reduction of the efficiency, as most likely due to a process involving polysulfide dissolution from the electrode into the electrolyte and precipitation at the anode side, with a significant loss of the active material and resistance increase. Apparently, the tregdme-based electrolyte mitigates neither the polysulfide dissolution nor the polysulfide shuttle, thereby leading to fast capacity fading and large irreversible capacity. A carbon-cloth support (i. e., a gas diffusion layer, GDL) is used to improve the stability. The microporous texture of the carbon cloth and the optimal electronic contact with the active material as well as the favorable chemical composition and wetting ability bring a large improvement in the cycling stability. This report agrees well with our result, which proves that the 3D

carbon matrix with identical merits with GDL is favorable to achieve an excellent performance.

Figure R2. Voltage profiles (a) and cycling trend with Coulombic efficiency (b) of a Na/TREGDME-NaCF₃SO₃/S-MWCNTs cell galvanostatically studied at a current of C/20 (1C = 1675 mA g⁻¹). Voltage limits 0.5 - 2.1 V. Charge capacity limited to 500 mAh g⁻¹. SMWCNTs cast on Al support. (c) Cyclic voltammetry of a Na/TREGDME-NaCF₃SO₃/S-MWCNTs cell and (d) related Nyquist plots of EIS measurements performed at the OCV and after the first, second, and third voltammetry cycles. Potential limits 0.5 - 2.1 V vs Na/Na⁺. Scan rate 0.1 mV s⁻¹. EIS performed within 100 kHz - 0.1 Hz using AC signal amplitude of 10 mV. S-MWCNTs cast on Al support. (e) Comparison of cycling trends and voltage profiles of the 25th cycle at C/20 (f) of two sodium cells using the TREGDME-NaCF₃SO₃ electrolyte and the SMWCNTs working electrode cast on conventional Al and gas diffusion layer (GDL) supports. Voltage limits 0.5 - 2.1 V for aluminum, 1.6 - 2.5 V for GDL. Charge

capacity limited to 500 mAh g⁻¹. Inset of panel f: voltage profiles of a comparative sodium cell using the TREGDME-NaCF₃SO₃ electrolyte and MWCNTs over GDL as working electrode; cell tested by employing the same experimental conditions of the Na/SMWCNTs (GDL) cell. Room temperature (25 °C) [*Ionics* **25**, 3129-3141,(2019).]

Updates to the revised manuscript: We have added the importance of the electrolyte in the revised manuscript on **Page 22**, highlighted in yellow.

Comment 5: Since the sulfurization process is carried out at the temperature of 155 °C and are not heat treated to remove surface sulfur molecules, there is a possibility of more sulfur anchored onto the surface of the carbon matrix. The presence of surface sulfur will result in polysulfide shuttling. Evidence to be provided to confirm the sulfur is fully confined inside the carbon matrix rather than on the surface.

Response: Thanks very much for the reviewer's professional and valuable comments. We would like to point out that the composite sample was heated to 200 °C for another 30 minutes to remove surface sulfur molecules. We have added the detailed process in the revised manuscript. FESEM images and EDS mapping images of S@BC-Ar, S@BC-NH₃ and S@Fe₃N-NMCN are provided in Figure 1k and Figure S12. It is obvious that three composites contain uniform distribution of sulfur without any bulk particles even under such a high S loading. The mapping images further confirm the uniform distribution of sulfur and also reveal that the self-standing structure is well preserved after S deposition.

Figure S12. FESEM and corresponding EDS mapping images of S@BC-Ar, S@BC-NH₃ and S@Fe₃N-NMCN.

Updates to the revised manuscript: We have added these details in the revised manuscript on Page 12, highlighted in yellow.

Comment 6: The morphology of S@Fe₃N-NMCN, S@BC-NH₃ and S@BC-Ar needs to be compared in order to understand the distribution of sulfur in the composite.

Response: Thank you very much for this valuable comment. In response to above **Comment 5**, we have further conducted the FESEM and EDS mapping analyses of S@BC-Ar, S@BC-NH₃ and S@Fe₃N-NMCN. The results demonstrate that three composites contain uniform distribution of sulfur without any bulk particles even under such a high S loading.

Reviewer #2

In the manuscript titled “Highly-efficient Fe₃N catalyst on N-doped hierarchical nanocarbons enables strong adsorption and fast dissociation of sodium polysulfides for advanced Na-S batteries”, the authors reported Fe₃N catalyst on N-doped hierarchical nanocarbons for room temperature Na-S battery. The Fe₃N catalyst shows affinity towards sodium polysulfides via Na-N and Fe-S bonds, which result in a better Na-S cycling. Overall, the work is interesting and is relevant for the battery community. However, the manuscript lacks in few parts, which need to be further elaborated and discussed. Therefore, I cannot recommend for publication in Nature Communications in the current format. A major revision is recommended for reconsideration. Below are my specific questions and comments.

Response: We thank the reviewer for the positive comments on the impact of our work. We have addressed these comments item by item and comprehensively improved our manuscript. Please refer to the highlighted part in the revised manuscript and supporting information.

Comment 1: The Fe₃N-NMCN was prepared in two steps, first at 300 °C for 1h and the second at 800 °C for 2h under NH₃ flow. I am curious why the synthesis proceeds in two steps? What is the mechanism for the carbonization of the bacterial cellulose and the growth of Fe₃N in NH₃ atmosphere? What is happening in the lower temperature and what at higher temperature? The reviewer suggests elaborating this part and to add a paragraph in the results and discussion to address these questions.

Response: Thanks for your valuable comments. The purpose of the treatment at the lower temperature of 300 °C is to stabilize the structure by evaporating those volatile species in the BC precursors (such as CO, CO₂, methanol and acetic acid)²⁶, before the final carbonization of BC at a higher temperature of 800 °C. Generally, to enable a better carbon design, a typical two-step process has been widely used to carbonize the precursors^{27,28} and other polymers^{29,30}.

To get insights on the carbonization mechanism of the bacterial cellulose, the microstructure and morphology of the product carbonized at different processes (the pristine BC, 300 °C 1h, 300 °C 1h and 500 °C 2h, 300 °C 1h and 800 °C 2h; abbreviated as BC-P, BC-300, BC-500 and BC-800) were characterized by XRD, Raman, XPS, FESEM and Elemental analysis, as summarized in Figure S1 and Tables S1-S2. The bare BC precursor shows three diffraction peaks at 14.3°, 16.7° and 22.5°. A broad peak of amorphous carbon appears at 21.5° after pyrolyzed at 300 °C and maintains almost unchanged with temperature increasing. Similarly, the Raman spectra of BC-300, BC-500 and BC-800 are almost the same. FESEM images in Figure S1d show that these composite fibers become thinner gradually with increasing temperatures.

Unlike the minor discrepancy in XRD and Raman, the surface functional groups on BC-P, BC-300, BC-500 and BC-800 are totally different. The C1s spectrum of the BC

precursor can be fitted into three peaks of C-C/C-H, C-OH and C(O)-O at binding energies of 284.8, 286.7 and 288.1 eV. Except for these three peaks, both BC-300, BC-500 and BC-800 possess strong bonds ascribed to C-N groups at around 285.4 eV. It is obvious that the proportion of C-N bonds compared with C-C/C-H declines as the increase of temperatures. In the O1s spectrum, three bands of O-C, O=C and O-C-O are detected at 530.7, 531.9 and 533.7 eV for BC-300. As the rise of temperature, the amount of O=C increases while that of O-C and O-C-O decreases dramatically. Finally, the O=C groups occupy about 63.4wt.% in BC-800, becoming the main part of oxygen. It has been reported that O=C groups can boost the affinity between the carbon anode and sodium ions, which is extremely favorable for performance enhancement^{29,30}. A weak N1s signal is observed for the pristine BC even before NH₃ treatment. The small amount of “intrinsic” nitrogen atoms come from the residual nitrogen-containing compounds left by the culture media and secretions. At 300 °C, pyridinic and pyrrolic N are two main N functional groups. As temperature rises, part of the pyridinic N transforms into pyrrolic, graphitic and oxidized N. Finally, the total content of pyridinic and pyrrolic N occupies 88.1% in BC-800, which can bond strongly to polysulfides and increase electronic conductivity. In a whole, as concluded in Table S2, the content of N/O is the highest at 300 °C and it decreases with the rise of temperature.

To accurately examine the variation of elemental content, elemental analysis was employed. As listed in Table S1, the pristine BC consists of 46.86wt.% C, 0.12wt.% N, 46.37wt.% O and 6.65wt.% H. Apparently, a small amount (0.12wt.%) of nitrogen is existed in BC-P, in line with the XPS result. When treated at 300 °C, most of the O/H volatilizes and a large amount of N (31.45wt.%) is doped. As temperature increases, the amount of N/O/H reduces and the content of C increases remarkably. Finally, the C, N, O and H in BC-800 is 89.77wt.%, 6.45wt.%, 2.56wt.% and 1.22wt.%. The elemental content and variation trend versus temperatures obtained by elemental analysis are highly consistent with the above-mentioned XPS results. In a whole, at 800 °C, a carbon matrix containing a certain amount of O=C, pyridinic N, pyrrolic N with little H is obtained.

Figure S1. XRD (a), Raman (b), XPS (c) and FESEM images (d) of BC-P, BC-300, BC-500 and BC-800.

Table S1. Elemental analysis of BC-P, BC-300, BC-500 and BC-800.

Weight percentage (wt.%)	BC-P	BC-300	BC-500	BC-800
C	46.86	56.83	71.85	89.77
N	0.12	31.45	20.65	6.45
O	46.37	9.07	4.76	2.56
H	6.65	2.64	2.74	1.22

Table S2. XPS analysis of BC-P, BC-300, BC-500 and BC-800.

Weight percentage (wt.%)	BC-P	BC-300	BC-500	BC-800
C	56.07	48.26	63.37	84.12
N	0.36	40.90	26.02	7.63
O	43.57	10.84	10.61	8.25

To understand the mechanism for the growth of Fe_3N , $\text{FeCl}_3 \cdot 6\text{H}_2\text{O}$ were sintered under NH_3 atmosphere at different processes (300 °C 1h, 300 °C 1h and 500 °C 2h, 300 °C 1h and 800 °C 2h; abbreviated as Fe-300, Fe-500, Fe-800). XRD patterns for products obtained at different temperatures are shown in Figure S2a. A few FeCl_3 is in the starting material due to the loss of crystal water. After heated at 300 °C, most of the sample converts into Fe_2O_3 with a small proportion of unreacted $\text{FeCl}_3 \cdot 6\text{H}_2\text{O}$. The product obtained at 300 °C presents as large particles. In the case of 500 °C, the Fe_3N dominates while a small amount of Fe_2O_3 , FeCl_3 , $\text{FeCl}_3 \cdot 6\text{H}_2\text{O}$ could still be discovered. The product obtained at 500 °C is composed of several small particles. Finally, a porous structured Fe_3N is obtained at 800 °C. In a whole, the $\text{FeCl}_3 \cdot 6\text{H}_2\text{O}$ turns into Fe_2O_3 at the lower temperature range (around 300 °C) and then the Fe_2O_3 fully converts into Fe_3N at 800 °C.

Figure S2. XRD patterns (a) and FESEM images (b-d) of the starting material, Fe-300, Fe-500 and Fe-800.

Updates to the revised manuscript: We have added these details in the revised manuscript on **Pages 6-7**, highlighted in yellow.

Comment 2: In the results and discussion only a vague and general sentence (“moreover, the composite possesses a mixture of micropores ($D < 2$ nm), mesopores ($2 \text{ nm} < D < 50$ nm) and macropores (inset of Figure 1f) is presented about the pores. To have a better picture of the materials ($\text{Fe}_3\text{N-NMCN}$, BC-NH_3 and BC-Ar), please provide the total pore volume (TPV at $P/P_0 = 0.99$), pore volume for the micro and pore volume for meso. Which DFT model was used to calculate the dV/dD vs. pore diameter? Does the surface area differ between the $\text{Fe}_3\text{N-NMCN}$, BC-NH_3 and BC-Ar ? Was the BET measured with N_2 or other gases? Please provide the information in the experimental part.

Response: Thank you very much for these good suggestions. The data for $\text{Fe}_3\text{N-NMCN}$ in the previous manuscript is obtained on Quantachrome Instruments, on which mesopores are the focus of concern. To have a better understanding of the pore-size distributions, BET testing through micromeritics ASAP2020 equipped with V4.02 software was employed for the analysis of total pores. The Non-Local Density Functional Theory (NLDFT) model was used to calculate the dV/dD vs. pore diameter curve. The BET was measured with N_2 . The pretreatment was applied at 120 °C for 24 h.

The adsorption/desorption isotherms and pore-size distribution curves of BC-Ar , BC-NH_3 and $\text{Fe}_3\text{N-NMCN}$ are displayed in Figure S6. The BET surface area, total pore volume, micropores volume and mesopores volume are summarized in Table S3. As displayed, the $\text{Fe}_3\text{N-NMCN}$ sample shows the highest BET surface area of $534.515 \text{ m}^2 \text{ g}^{-1}$ while those of BC-Ar and BC-NH_3 are 410.780 and $425.128 \text{ m}^2 \text{ g}^{-1}$.

The total pore volume, micropores volume and mesopores volume for $\text{Fe}_3\text{N-NMCN}$ are 0.55280 , 0.17226 , $0.38054 \text{ cm}^3 \text{ g}^{-1}$, respectively. The ratio of micropores is 31.2%. In the case of BC-NH_3 and BC-Ar , both the total pore volume and the ratio of micropores are reduced. It is likely that the heat treatment in NH_3 and loading of Fe_3N can introduce some mesopores and then enhance surface areas, which is beneficial for the immobilization of sulfur thus enabling a high-loading cathode.

Table S3. BET surface areas, total pore volume, micropores volume and mesopores volume of BC-Ar , BC-NH_3 and $\text{Fe}_3\text{N-NMCN}$.

Sample	Surface area ($\text{m}^2 \text{g}^{-1}$)	Total pore volume ($\text{cm}^3 \text{g}^{-1}$)	Micropores volume ($\text{cm}^3 \text{g}^{-1}$)	Mesopores volume ($\text{cm}^3 \text{g}^{-1}$)	Ratio of micropores (%)
BC-Ar	410.780	0.40613	0.11672	0.28941	28.8
BC-NH ₃	425.128	0.43570	0.12818	0.30752	29.4
Fe ₃ N-NMCN	534.515	0.55280	0.17226	0.38054	31.2

Figure S6. Adsorption/desorption isotherms and pore-size distribution curves of BC-Ar (a-b), BC-NH₃ (c-d) and Fe₃N-NMCN (e-f).

Updates to the revised manuscript: We have added these details in the revised manuscript on **Page 7**, highlighted in yellow.

Comment 3: The authors claim to have the N doping of the carbon materials, which is supported by EDS and XPS. However, there is no amount provided. How much of the N doping of the carbon is present in the Fe₃N-NMCN and BC-NH₃? Moreover, also the Fe₃N amount is missing. Please provide the amounts in wt.%. Are the N-doping and Fe₃N present on carbon surface or are embedded in the bulk of the material?

Response: Thanks very much for the constructive comment.

(1) Fe₃N amount in the Fe₃N-NMCN composite.

TGA tests were employed to detect the loading of Fe₃N in the Fe₃N-NMCN composite. To clarify the transformation process of Fe₃N, TGA tests of the bare Fe₃N powder and Fe₃N-NMCN in O₂ atmosphere were conducted (Figure S8a-b). Moreover, the phase information of the final products after TGA tests was also detected by XRD (Figure S8c-d). The bare Fe₃N powder was obtained through the heat treatment of FeCl₃·6H₂O at 300 °C for 1 h and then 800 °C for 2 h.

As displayed in Figure S8c-d, both products obtained after TGA tests are Fe₂O₃ (159.6 g mol⁻¹). When we assume that there is 1 mol Fe₃N (181.4 g mol⁻¹), there should be 1.5 mol Fe₂O₃ after the TGA test since the mole number of Fe is constant. During this process, the weight increment is 132.0wt.% (=159.6*1.5/181.4). This result is highly consistent with the TGA curve of the bare Fe₃N powder in Figure S8a, where the weight increase between room temperature and 800 °C is about 134wt.% and the main weight variation is in the range of 350 and 800 °C.

Therefore, it can be deduced that the weight loss (89wt.%) of Fe₃N-NMCN before 353 °C (Figure S8b) is exclusively originated from the consumption of carbon matrix in Fe₃N-NMCN and is not related to Fe₃N, while the weight increment (132wt.%=14.5wt.%/11wt.%) between 353 and 800 °C is only from the oxidation of Fe₃N to Fe₂O₃. From above analyses, it is determined that the weight percentage of Fe₃N in Fe₃N-NMCN is 11wt%.

Figure S8. Calculation of Fe₃N content in the Fe₃N-NMCN composite.

(a-b) TGA tests of the bare Fe₃N powder (a) and the Fe₃N-NMCN composite (b).

(c-d) XRD patterns of the final products after TGA tests from the bare Fe₃N powder (c) and the Fe₃N-NMCN composite (d).

(2) Content of doped N in these composites.

Elemental analysis (Table S4) was conducted to obtain the total content of C, N, H and O in these composite. As demonstrated, the doped N in BC-NH₃ and BC-Ar are 6.45wt.% and 0.34wt.%, respectively. Note that the small amount of “intrinsic” nitrogen atoms in BC-Ar come from the residual nitrogen-containing compounds left by the culture media and secretions. In addition, Table S4 shows that the weight ratio of C, N, H, O in the Fe₃N-NMCN are 61.50wt.%, 10.21wt.%, 1.48wt.% and 12.38wt.%, respectively. According to the above TGA result, the loading of Fe₃N is 11wt.% and the N from the Fe₃N is calculated to 0.85wt.% ($=11 \times 14 / (56 \times 3 + 14) = 0.85 \text{ wt.} \%$). Thus, the doped N in the composite is 9.36wt.% ($10.21 - 0.85 = 9.36 \text{ wt.} \%$). In a whole, the N doped in BC-Ar, BC-NH₃ and Fe₃N-NMCN are 0.34wt.%, 6.45wt.% and 9.36wt.%, respectively.

Table S4. Elemental analysis of BC-Ar, BC-NH₃ and Fe₃N-NMCN.

Weight percentage (wt.%)	BC-Ar	BC-NH ₃	Fe ₃ N-NMCN
C	90.70	89.77	61.50
H	1.70	1.22	1.48
N	0.34	6.45	10.21
O	7.26	2.56	12.38

(3) The TEM images in Figure 1d-e and Figure S5 show the Fe₃N quantum dots are coated with carbon layers. To further detect whether the N-doping and Fe₃N present on fiber surface or are embedded in the fiber, XPS depth profiling results after 0, 60, 120, 180 s of sputtering were collected as Figure S9. As displayed, all elements (C, O, N, Fe) are existed throughout the sample. Moreover, the shape and intensity of peaks maintain almost unchanged. The above results further demonstrate that both the N-doping and Fe₃N are uniformly distributed through the whole sample.

Figure S9. XPS depth profiles after 0, 60, 120, 180 s of sputtering.

Updates to the revised manuscript: We have added these details in the revised manuscript on Page 11, highlighted in yellow.

Comment 4: The authors claim that Fe₃N has “superior electronic conductivity and high ionic diffusion ability”. To what is these compared? How much are these values? Are these conclusions supported by EIS?

Response: Thanks for your valuable comment. We would like to indicate that usually metal nitrides have better electronic conductivity and ionic diffusion ability than

metal oxides³¹. It has been reported that Fe₃N exhibits a high electronic conductivity of 0.1-1.0×10⁻⁵ S cm⁻¹³² and an ion diffusion coefficient of 1.74-3.56×10⁻¹³ cm² s⁻¹³³. However, the electronic conductivity and ion diffusion coefficient of Fe₂O₃ are only 10⁻¹⁴ S cm⁻¹³⁴ and 10⁻¹⁴-10⁻¹⁵ cm² s⁻¹³⁵.

The EIS results (Figure 5d and Figure S22, Tables S6-S7) agree well with this claim. Before cycle, the battery with S@Fe₃N-NMCN shows the lowest resistance (67.5 Ω) compared to S@BC-NH₃ (93.9 Ω) and S@BC-Ar (119.7 Ω). Moreover, the battery with S@Fe₃N-NMCN displays the largest slope at the low frequency Warburg diffusion range, indicating the fastest diffusion of ions in the S@Fe₃N-NMCN electrode. Furthermore, the battery with S@Fe₃N-NMCN exhibits the lowest SEI impedance (2.03 Ω) and charge transfer impedance (1.40 Ω) after cycle. The enhancement in ionic diffusion and charge transfer behavior of S@Fe₃N-NMCN compared to their counterparts can be attributed to the excellent ionic diffusion and electronic conductivity of Fe₃N. This further certifies the excellent rate capability and also agrees well with the claim that Fe₃N has superior electronic conductivity and high ionic diffusion ability.

Figure S22. EIS spectra and corresponding equivalent circuit diagrams of S@BC-Ar, S@BC-NH₃, S@Fe₃N-NMCN before (a, c) and after cycle (b, d).

Table S6. Fitted EIS results of batteries with S@BC-Ar, S@BC-NH₃ and S@Fe₃N-NMCN before cycle.

Sample	R _s (Ω)	R _p (Ω)
--------	--------------------	--------------------

S@Fe ₃ N-NMCN	3.97	67.46
S@BC-NH ₃	0.81	93.88
S@BC-Ar	1.39	119.70

Table S7. Fitted EIS results of batteries with S@BC-Ar, S@BC-NH₃ and S@Fe₃N-NMCN after cycle.

Sample	R _s (Ω)	R _f (Ω)	R _{ct} (Ω)
S@Fe ₃ N-NMCN	7.02	2.03	1.40
S@BC-NH ₃	3.60	2.10	1.53
S@BC-Ar	4.71	6.21	1.92

Updates to the revised manuscript: We have added the details in the revised manuscript on Pages 24-25, highlighted in yellow.

Comment 5: What was the Na₂S₆ concentration in the solution?

Response: The concentration of the Na₂S₆ is 0.1 mol L⁻¹. The Na₂S₆ solution (0.1 mol L⁻¹) was prepared by mixing sodium sulfide (Na₂S) and sulfur with a molar ratio of 1:5 in 1,3-dioxolane (DOL) and Diethylene glycol dimethyl ether (DIGLYME) (1 : 1 Vol.%). Then, the solution was sealed with inert gas and stirred at room temperature for 20 h. 5 mg Fe₃N-NMCN, BC-NH₃, BC-Ar, GO and super P were added into the diluted Na₂S₆ solution, respectively, with the blank Na₂S₆ solution as a reference.

Updates to the revised manuscript: We have added the detail in the revised manuscript on Page 34, highlighted in yellow.

Comment 6: Did the authors perform a post-mortem analysis after prolong cycling? What was the separator color? How is look like the Na metal anode? The reviewer is curious what happened to the Na metal anode at a current density of 5C after prolong cycling and at 10C? Do Na dendrites form? How much is the Na anode preserved?

Response: Thank you for your valuable questions. Post-mortem analyses for the separators, the Na metal anodes and the as-employed electrodes disassembled from batteries with S@BC-Ar, S@BC-NH₃ and S@Fe₃N-NMCN were also conducted, as shown in Figures S29-S34. Generally, serious dissolution and shuttle of polysulfides

lead to more yellowish separators. As can be seen, in contrast to the most yellowish separator from S@BC-Ar, the separator from S@BC-NH₃ shows less yellowish and no visible yellow color can be observed in the separator from S@Fe₃N-NMCN, suggesting increasingly inhibited dissolution and shuttle of polysulfides. For the cycled Na electrodes, dendrites are barely found in case of S@Fe₃N-NMCN when compared to their counterparts. In addition, a loose layer is formed on the cycled Na anode from BC-Ar, while the surface with S@Fe₃N-NMCN is smooth and clean due to the less shuttling of polysulfides. Figures S31-S34 show that all three cycled electrodes can maintain the original morphology and all elements are uniformly distributed without aggregation; however, the S@BC-Ar electrode shows a smoother surface, indicating the dissolution of some active sulfur materials into the electrolyte. The above results thus suggest that the dissolution and shuttle of polysulfides can be largely inhibited by Fe₃N-NMCN.

Figure S29. Optical images of the separators from batteries with S@BC-Ar, S@BC-NH₃ and S@Fe₃N-NMCN, respectively.

Figure S30. FESEM images of cycled Na metal paired with S@BC-Ar, S@BC-NH₃ and S@Fe₃N-NMCN, respectively.

Figure S31. FESEM images of cycled S@BC-Ar, S@BC-NH₃ and S@Fe₃N-NMCN electrodes, respectively.

Figure S32. EDS mapping images of cycled S@BC-Ar.

Figure S33. EDS mapping images of cycled S@BC-NH₃.

Figure S34. EDS mapping images of cycled S@Fe₃N-NMCN.

Updates to the revised manuscript: We have added the detail in the revised manuscript on Pages 27-28, highlighted in yellow.

Comment 7: Coulombic efficiency and capacity vs. cycle graphs should use reasonable y-axis scales. For instance, Coulombic efficiency should not be reported on a y-axis scale of 0-100% but rather 90-100%.

Response: Thanks for your constructive suggestion. We have changed the y-axis scale of Coulombic efficiency, as shown in the revised manuscript.

Comment 8: The authors claim that “It should be emphasized that even under the identical preparation procedure (mass ratio of S, temperature, etc.), smaller amounts of sulfur were immobilized in S@BC-NH₃ (74%) and S@BC-Ar (62%) compared with S@Fe₃N-NMCN (85%). These results are well consistent with the adsorption test in Figure 2a, where the adsorption ability of these substrates follows an order of Fe₃N-NMCN, BC-NH₃ and BC-Ar.” Is the melt infiltration of sulfur and the polysulfides adsorption two different physical phenomena? The melt infiltration is related to the filling of the total pore volume of the material. On the other hand, the polysulfides adsorption is the interplay the physical-chemical properties of the materials (different doping, surface area, porosity). As it was asked in the question 2, the total pore amount for all the materials should be provided to have a better comparison.

Response: Thanks for your professional comments to improve our manuscript.

Indeed, the melt infiltration is related to the filling of the total pore volume of the material but the polysulfides adsorption is the interplay of the physical-chemical properties of the materials (different doping, surface area, porosity). We have elaborated the polysulfides adsorption result again, from aspects of doping, surface area and porosity. In a whole, the Fe₃N-NMCN sample displays the highest N-doping (9.36wt.% vs. 6.45wt.% for BC-NH₃ and 0.34wt.% for BC-Ar), surface area (534.515 m² g⁻¹ vs. 425.128 m² g⁻¹ for BC-NH₃ and 410.780 m² g⁻¹ for BC-Ar) and pore

volumes ($0.55280 \text{ cm}^3 \text{ g}^{-1}$ vs. $0.43570 \text{ cm}^3 \text{ g}^{-1}$ for BC-NH₃ and $0.40613 \text{ cm}^3 \text{ g}^{-1}$ for BC-Ar) as well as 11wt.% Fe₃N. In this way, the adsorption ability of these substrates follows a decreasing order of Fe₃N-NMCN, BC-NH₃ and BC-Ar.

(1) N-doping

Elemental analysis (Table S4) was conducted to obtain the total content of C, N, H and O in these composites. As demonstrated, the doped N in BC-NH₃ and BC-Ar are 6.45wt.% and 0.34wt.%, respectively. Note that the small amount of “intrinsic” nitrogen atoms in BC-Ar come from the residual nitrogen-containing compounds left by the culture media and secretions. In addition, Table S4 shows that the weight ratio of C, N, H, O in the Fe₃N-NMCN are 61.50wt.%, 10.21wt.%, 1.48wt.% and 12.38wt.%, respectively. According to the above TGA result, the loading of Fe₃N is 11wt.% and the N from the Fe₃N is calculated to 0.85wt.% ($=11 \times 14 / (56 \times 3 + 14) = 0.85 \text{ wt.} \%$). Thus, the doped N in the composite is 9.36wt.% ($=10.21 - 0.85 = 9.36 \text{ wt.} \%$). In a whole, the N doped in BC-Ar, BC-NH₃ and Fe₃N-NMCN are 0.34wt.%, 6.45wt.% and 9.36wt.%, respectively.

Table S4. Elemental analysis of BC-Ar, BC-NH₃ and Fe₃N-NMCN.

Weight percentage (wt.%)	BC-Ar	BC-NH ₃	Fe ₃ N-NMCN
C	90.70	89.77	61.50
H	1.70	1.22	1.48
N	0.34	6.45	10.21
O	7.26	2.56	12.38

(2) Surface area and porosity

The adsorption/desorption isotherms and pore-size distribution curves of BC-Ar, BC-NH₃ and Fe₃N-NMCN are displayed in Figure S6. The BET surface area, total pore volume, micropores volume and mesopores volume are summarized in Table S3. As displayed, the Fe₃N-NMCN sample shows the highest BET surface area of $534.515 \text{ m}^2 \text{ g}^{-1}$ while those of BC-Ar and BC-NH₃ are 410.780 and $425.128 \text{ m}^2 \text{ g}^{-1}$.

The total pore volume, micropores volume and mesopores volume for Fe₃N-NMCN are 0.55280 , 0.17226 , $0.38054 \text{ cm}^3 \text{ g}^{-1}$, respectively. The ratio of micropores is 31.2%. In the case of BC-NH₃ and BC-Ar, both the total pore volume and the ratio of micropores are reduced. It is likely that the heat treatment in NH₃ and loading of Fe₃N can introduce some mesopores and then enhance surface areas, which is beneficial for the immobilization of sulfur thus enabling a high-loading cathode.

Table S3. BET surface areas, total pore volume, micropores volume and mesopores volume of BC-Ar, BC-NH₃ and Fe₃N-NMCN.

Sample	Surface area (m ² g ⁻¹)	Total pore volume (cm ³ g ⁻¹)	Micropores volume (cm ³ g ⁻¹)	Mesopores volume (cm ³ g ⁻¹)	Ratio of micropores (%)
BC-Ar	410.780	0.40613	0.11672	0.28941	28.8
BC-NH ₃	425.128	0.43570	0.12818	0.30752	29.4
Fe ₃ N-NMCN	534.515	0.55280	0.17226	0.38054	31.2

Figure S6. Adsorption/desorption isotherms and pore-size distribution curves of BC-Ar (a-b), BC-NH₃ (c-d) and Fe₃N-NMCN (e-f).

Updates to the revised manuscript: We have added the detail in the revised manuscript on Page 14, highlighted in yellow.

Comment 9: In the experimental part a lot of information's are missing, such as:
I. Did the authors perform a purification step after the pyrolysis to remove some impurities? Please provide a sentence.

Response: Thanks for pointing out this detail. No purification step after the pyrolysis was employed in the synthesis of Fe₃N-NMCN. We have added the detail in the revised manuscript on Page 31, highlighted in yellow.

II. How was the Na₂S₆ prepared?

Response: The Na₂S₆ solution (0.1 mol L⁻¹) was prepared by mixing sodium sulfide (Na₂S) and sulfur with a molar ratio of 1:5 in 1,3-dioxolane (DOL) and Diethylene glycol dimethyl ether (DIGLYME) (1 : 1 Vol.%). Then, the solution was sealed with inert gas and stirred at room temperature for 20 h. 5 mg Fe₃N-NMCN, BC-NH₃, BC-Ar, GO and super P were added into the diluted Na₂S₆ solution respectively, with the blank Na₂S₆ solution as a reference. We have added the detail in the revised manuscript on Page 32, highlighted in yellow.

III. What was the typical sulfur loading on the electrode? Please provide the areal mass (mg/cm²) loading of the sulfur active material

Response: The areal mass loading of the sulfur active material in the Fe₃N-NMCN electrode is around 2.6 mg cm⁻². We have added the detail in the revised manuscript on Pages 31-32, highlighted in yellow.

IV. What was the cathode thickness?

Response: The thickness of the cathode is around 300 μm, as shown in Figure S39. We have added the detail in the revised manuscript on Page 31, highlighted in yellow.

Figure S39. Side view FESEM image of S@Fe₃N-NMCN.

V. For the XPS measurements. Was any inert transfer for the materials which were in contact with the polysulfides?

Response: Firstly, these materials were filtrated, dried and collected in the Ar glove box. Secondly, these materials were pasted on the XPS platform and the platform was sealed with a plastic centrifuge tube. Thirdly, the platform was transferred into the XPS chamber and vacuumed quickly. We have added the detail in the revised manuscript on Pages 33-34, highlighted in yellow.

VI. How the in-situ Raman cell look like? Please provide a schematics or reference.

Response: The in situ Raman cell is purchased from EL-CELL company. The schematics of the in situ Raman cell are shown in Figure S40. We have added the detail in the revised manuscript on Page 33, highlighted in yellow.

Figure S40. Schematics of the in situ Raman cell.

VII. What was the power of the 512 nm laser and how much was the acquisition time per spectra? For the in-situ Raman, a galvanostatic graph for the in-situ discharge and charge should be provided.

Response: We are sorry for the typo error. The wavenumber of the Raman laser is 532 nm and the power of the 532 nm laser is 50 mW. The acquisition time per spectra is 24 second. The discharge/charge process of the in situ Raman is performed by a cyclic voltammetry program on CHI760E. The CV curve has been added as Figure S28.

Figure S28. The CV curve for *in situ* Raman.

VII. What was the sodium foil thickness? Was any pretreatment of the metal anode before the battery assembly?

Response: The thickness of the sodium foil is around 600 μm , as displayed in Figure S41. The Na metal was used directly without any pretreatment.

Figure S41. Side view FESEM images of sodium foils.

VIII. To have a better comparison with the literature, please provide the normalized sulfur electrolyte volume ratio (the E/S in $\mu\text{L}/\text{mg S}$) not only the volume per cell.

Response: Thanks for the comment. The normalized sulfur electrolyte volume ratio (the E/S ratio) is 18.2 $\mu\text{L mg}^{-1}$. The weight of the $\text{Fe}_3\text{N-NMCN}$ support is ~ 1 mg, while the weight of $\text{S@Fe}_3\text{N-NMCN}$ is ~ 6.5 mg. The sulfur in the $\text{S@Fe}_3\text{N-NMCN}$

electrode is ~ 5.5 mg. The diameter of S@Fe₃N-NMCN is ~ 16 μ m. The electrolyte/sulfur (E/S) ratio is 18.2 μ L mg⁻¹ (100 μ L/5.5 mg=18.2 μ L mg⁻¹).

IX. In the first-principles calculation/simulation experimental part please use the correct referencing. All the references from the first-principles calculation/simulation should be put in the reference section.

Response: Thanks. We have put references of the first-principles calculation/simulation in the reference section, as Refs. 41-45 in the revised manuscript.

Minor issues:

1. There are some vague claims which are not supported by the data or references, such as “excellent electronic conductivity”.

Response: Thanks for the good suggestion. Yu³⁶ et al. claimed that the electronic conductivity of carbonized bacterial cellulose aerogel reaches 20.6 S m⁻¹. In addition, it has been reported by Xie³³ that Fe₃N exhibits a high electronic conductivity of 0.1-1.0 $\times 10^{-5}$ S cm⁻¹. We have cited these references (Refs. 26, 36, 37) in the revised manuscript.

2. Please tone down the word superior, it is used to frequently in the manuscript.

Response: We have tone down the word “superior”. Please refer to the revised manuscript.

3. The authors showed an example of Se as other multi-electron redox materials. The reviewer is curious if Fe₃N-NMCN can be efficiently used also in Li-S and Mg-S batteries?

Response: To demonstrate the effectiveness, the S@Fe₃N-NMCN electrode is also used for Li-S batteries. Figure S38a shows that the S@Fe₃N-NMCN electrode can deliver a high reversible capacity of 1345.7 mA h g⁻¹ with an ICE of 94.3% at 167.5 mA g⁻¹, when 1M LiTFSI in DOL/DME (1:1 by volume) with 0.1 M LiNO₃ was applied as the electrolyte. However, it should be noted that the ester-based electrolyte is not suitable for the S@Fe₃N-NMCN electrode of Li-S batteries (Figure S38b), as the same case in Na-S batteries.

Figure S38. Discharge/charge curves of S@Fe₃N-NMCN with 1M LiTFSI in DOL/DME (1:1 by volume) containing 0.1M LiNO₃ additive (a) and 1M LiPF₆ in EC/DEC/EMC (1:1:1 by volume) electrolyte (b) in Li-S batteries.

Reviewer #3

This work reports an N-doped self-standing electrode using a Fe₃N catalyst for room-temperature (RT) Na-S batteries. The authors provided various experimental data on a high-performance Na-S cell benefiting from the proposed electrode composition, along with adsorption tests and simulations to investigate the catalytic effect of Fe₃N. The high capacity, cycling stability, and rate capability shown in this report might be of interest for the scientific community working in the field of RT Na-S batteries. However, the results provided by the authors do not clearly elucidate the actual role of Fe₃N on the electrochemical behavior of Na-S batteries, and the conclusions appear to be not properly supported by evidence. Therefore, we suggest resubmission to a more specialized journal after addressing the major issues listed below.

Response: Thank you very much for your helpful comments. According to your suggestions, we have conducted a huge number of new experiments to elucidate the fundamental reasons underlying the electrochemical behavior and emphasized the novelty of our work, including discussion on the strategy to tackle the polysulfide dissolution issue, some experimental details especially about the composition of these electrodes, new Na-S cell testing experiments in lean electrolyte and post-mortem cell components characterizations, new in situ Raman with enhanced resolution and so on

To sum up, this work exhibits the following interesting and important points to the scientific community especially the readers of Nature Communications.

i) With a remarkable sulfur loading of 85wt.%, outstanding electrochemical performances up to 699 mA h g⁻¹ after 2800 cycles at a rather high current density of 8375 mA g⁻¹ in a common E/S ratio range is achieved, which is better than most of the recently published literatures on Metal-S batteries¹⁹⁻²². Furthermore, an acceptable capacity of 810.5 mA h g⁻¹ at 167.5 mA g⁻¹ is still obtained when **a very low E/S ratio of 7.27 μL mg⁻¹** was used.

ii) Experimental and theoretical results reveal that Fe₃N shows prominent affinity to sodium polysulfides via Na-N and Fe-S bonds, which **makes NaPSs dissociated rapidly**. A highly reversible reaction between Na₂S₈ and Na₂S is observed by in situ Raman in a common E/S ratio range.

iii) The proposed electrode can also work well in Li-S and Na-Se batteries.

We believe this paper offers a fundamental understanding of the Fe₃N catalyst mechanism and opens a new avenue to design catalyst material for high-loading, fast-kinetics, polysulfides-retention Metal-S batteries. For these reasons, we believe the revised manuscript should be of great interest to researchers in materials science and engineering, electrochemistry, nanoscience and nanotechnology and energy technology.

Comment 1: The authors reported a sulfur loading on the cathode of 2.6 mg/cm^2 , which may be considered rather promising, although further improvements are certainly needed to enable practical applications (Joule 4, 285–291 (2020)). On the other hand, other crucial electrode metrics are missing, such as thickness and weight of the $\text{Fe}_3\text{N-NMCN}$ support, as well as the electrolyte/sulfur (E/S) ratio. These parameters are needed for a proper evaluation of the cell performance, in particular considering the non-conventional, self-standing cathode configuration, and should be compared to those of the benchmark electrodes shown in Figs. S9, S11, S12. A lack of microstructural and cell metric data on both the proposed and the benchmark electrodes and does not allow to separate the effects of cathode morphology and Fe_3N catalyst.

Response: Thanks the reviewer very much for valuable comments. After reading this valuable Joule paper, we have conducted a series of new experiments to provide more details in the revised manuscript. This paper was cited as Ref. 18.

(1) Crucial electrode metrics. The thickness of the $\text{Fe}_3\text{N-NMCN}$ matrix is around $350 \text{ }\mu\text{m}$, as demonstrated by the side view FESEM images in Figure S4. The weight of the $\text{Fe}_3\text{N-NMCN}$ support is $\sim 1 \text{ mg}$, while the weight of $\text{S@Fe}_3\text{N-NMCN}$ is $\sim 6.5 \text{ mg}$. The sulfur in the $\text{S@Fe}_3\text{N-NMCN}$ electrode is $\sim 5.5 \text{ mg}$. The diameter of $\text{S@Fe}_3\text{N-NMCN}$ is $\sim 16 \text{ mm}$. The electrolyte/sulfur (E/S) ratio is $18.2 \text{ }\mu\text{L mg}^{-1}$ ($100 \text{ uL}/5.5 \text{ mg}=18.2 \text{ }\mu\text{L mg}^{-1}$). The weight of the BC- NH_3 and BC-Ar support is $\sim 0.85 \text{ mg}$, while the weight of S@BC-NH_3 and S@BC-Ar are ~ 3.2 and 2.3 mg . The sulfur in S@BC-NH_3 and S@BC-Ar are ~ 2.35 and 1.45 mg . The electrolyte/sulfur (E/S) ratio for S@BC-NH_3 and S@BC-Ar is 42.6 and $69.0 \text{ }\mu\text{L mg}^{-1}$. These details are added in the experimental part in the revised manuscript on Pages 31-32 and supporting information.

Figure S4. Side view FESEM image of $\text{Fe}_3\text{N-NMCN}$.

(2) Microstructural data on both the proposed and the benchmark electrodes.

FESEM images and EDS mapping images of S@BC-Ar, S@BC-NH₃ and S@Fe₃N-NMCN are compared in Figure 1k and Figure S12. It is obvious that three composites contain uniform distribution of sulfur without any bulk particles even under such a high S loading. The mapping images further confirm the uniform distribution of sulfur and also reveal that the self-standing structure is well preserved after S deposition. In addition, there is little difference in the cathode morphology between S@BC-Ar, S@BC-NH₃ and S@Fe₃N-NMCN.

Figure S12. FESEM and corresponding EDS mapping images of S@BC-Ar, S@BC-NH₃ and S@Fe₃N-NMCN.

(3) However, there is much distinction in doping, surface area and pore structure between BC-Ar, BC-NH₃ and Fe₃N-NMCN.

i. N-doping

Elemental analysis (Table S4) was conducted to obtain the total content of C, N, H and O in these composites. As demonstrated, the doped N in BC-NH₃ and BC-Ar are 6.45wt.% and 0.34wt.%, respectively. Note that the small amount of “intrinsic” nitrogen atoms in BC-Ar come from the residual nitrogen-containing compounds left by the culture media and secretions. In addition, Table S4 shows that the weight ratio of C, N, H, O in the Fe₃N-NMCN are 61.50wt.%, 10.21wt.%, 1.48wt.% and 12.38wt.%, respectively. According to the above TGA result (See response for

Reviewer 1), the loading of Fe₃N is 11wt.% and the N from the Fe₃N is calculated to 0.85wt.% ($=11 \times 14 / (56 \times 3 + 14) = 0.85 \text{wt.}\%$). Thus, the doped N in the composite is 9.36wt.% ($10.21 - 0.85 = 9.36 \text{wt.}\%$). In a whole, the N doped in BC-Ar, BC-NH₃ and Fe₃N-NMCN are 0.34wt.%, 6.45wt.% and 9.36wt.%, respectively.

Table S4. Elemental analysis of BC-Ar, BC-NH₃ and Fe₃N-NMCN.

Weight percentage (wt.%)	BC-Ar	BC-NH ₃	Fe ₃ N-NMCN
C	90.70	89.77	61.50
H	1.70	1.22	1.48
N	0.34	6.45	10.21
O	7.26	2.56	12.38

ii. Surface area and porosity

The adsorption/desorption isotherms and pore-size distribution curves of BC-Ar, BC-NH₃ and Fe₃N-NMCN are displayed in Figure S6. The BET surface area, total pore volume, micropores volume and mesopores volume are summarized in Table S3. As displayed, the Fe₃N-NMCN sample shows the highest BET surface area of 534.515 m² g⁻¹ while those of BC-Ar and BC-NH₃ are 410.780 and 425.128 m² g⁻¹.

The total pore volume, micropores volume and mesopores volume for Fe₃N-NMCN are 0.55280, 0.17226, 0.38054 cm³ g⁻¹, respectively. The ratio of micropores is 31.2%. In the case of BC-NH₃ and BC-Ar, both the total pore volume and the ratio of micropores are reduced. It is likely that the heat treatment in NH₃ and loading of Fe₃N can introduce some mesopores and then enhance surface areas, which is beneficial for the immobilization of sulfur thus enabling a high-loading cathode.

Table S3. BET surface areas, total pore volume, micropores volume and mesopores volume of BC-Ar, BC-NH₃ and Fe₃N-NMCN.

Sample	Surface area (m ² g ⁻¹)	Total pore volume (cm ³ g ⁻¹)	Micropores volume (cm ³ g ⁻¹)	Mesopores volume (cm ³ g ⁻¹)	Ratio of micropores (%)
BC-Ar	410.780	0.40613	0.11672	0.28941	28.8
BC-NH ₃	425.128	0.43570	0.12818	0.30752	29.4

Figure S6. Adsorption/desorption isotherms and pore-size distribution curves of BC-Ar (a-b), BC-NH₃ (c-d) and Fe₃N-NMCN (e-f).

Updates to the revised manuscript: We have added the detail in the revised manuscript on Pages 7, 14, 31-34, highlighted in yellow.

Comment 2: The authors should further investigate the reasons behind the observed change in UV/Vis spectra of the catholyte solutions when in contact with the several electrode powders (Fig. 2a–b). XRD and SEM-EDS analyses of these powders should

be performed to verify whether that change is due to adsorption or chemical reaction with precipitation of S-containing species. Moreover, XRD and SEM-EDS data of various electrodes after cycling in the cell should be provided to demonstrate the reversibility of the electrochemical reaction.

Response: Thanks very much for these good suggestions.

(1) To verify whether the observed change in UV-Vis spectra is caused by adsorption or chemical reaction with precipitation of S-containing species, we **analyzed the powders after 24 hours adsorption by XRD, SEM and EDS**. As shown in Figure S13a, the XRD patterns show no other peaks other than the (002) diffraction peak at $\sim 26.0^\circ$ (due to amorphous carbon) and the peaks corresponding to Fe_3N in $\text{Fe}_3\text{N-NMCN}$. This suggests that no new compounds form as a result of chemical reactions. Furthermore, SEM and EDS mapping images in Figure S13b-i demonstrate that there is no aggregation and all elements are uniformly dispersed in these substrates. The above results thus verify that the changes in UV-Vis spectra should be originated from the adsorption of Na_2S_6 .

Figure S13. XRD (a), FESEM images (b-e) and EDS mapping images (f-g) of $\text{Super P-Na}_2\text{S}_6$, $\text{BC-Ar-Na}_2\text{S}_6$, $\text{BC-NH}_3\text{-Na}_2\text{S}_6$ and $\text{Fe}_3\text{N-NMCN-Na}_2\text{S}_6$.

(2) **Post-mortem analyses** for the separators, the Na metal anodes and the as-employed electrodes disassembled from batteries with S@BC-Ar , S@BC-NH_3 and

S@Fe₃N-NMCN, respectively, were also conducted, as shown in Figures S29-S34. Generally, serious dissolution and shuttle of polysulfides lead to more yellowish separators. As can be seen, in contrast to the most yellowish separator from S@BC-Ar, the separator from S@BC-NH₃ shows less yellowish and no visible yellow color can be observed in the separator from S@Fe₃N-NMCN, suggesting increasingly inhibited dissolution and shuttle of polysulfides. For the cycled Na electrodes, dendrites are barely found in case of S@Fe₃N-NMCN when compared to their counterparts. In addition, a loose layer is formed on the cycled Na anode from BC-Ar, while the surface with S@Fe₃N-NMCN is smooth and clean due to the less shuttling of polysulfides. Figures S31-S34 show that all three cycled electrodes can maintain the original morphology and all elements are uniformly distributed without aggregation; however, the S@BC-Ar electrode shows a smoother surface, indicating the dissolution of some active sulfur materials into the electrolyte. The above results thus suggest that the dissolution and shuttle of polysulfides can be largely inhibited by Fe₃N-NMCN.

Figure S29. Optical images of the separators from batteries with S@BC-Ar, S@BC-NH₃ and S@Fe₃N-NMCN, respectively.

Figure S30. FESEM images of cycled Na metal paired with S@BC-Ar, S@BC-NH₃ and S@Fe₃N-NMCN, respectively.

Figure S31. FESEM images of cycled S@BC-Ar, S@BC-NH₃ and S@Fe₃N-NMCN electrodes, respectively.

Figure S32. EDS mapping images of cycled S@BC-Ar.

Figure S33. EDS mapping images of cycled S@BC-NH₃.

Figure S34. EDS mapping images of cycled S@Fe₃N-NMCN.

Updates to the revised manuscript: We have added the detail in the revised manuscript on Pages 27-28, highlighted in yellow.

Comment 3: The change in voltage profiles during the long-term cycling test should be shown (Fig. 5f). Does the voltage curve vary after 2800 cycles?

Response: Thanks for pointing out this detail. We have added the voltage-capacity curves during the long-term cycling test as Figure S25. It can be found that there is little change in the voltage curves after 2800 cycles. The long-term cycling test at 8375 mA g⁻¹ is tested after 10 cycles at 167.5 mA g⁻¹. This detail has been added in the experimental part.

Figure S25. Voltage-capacity curves during the long-term test.

Comment 4: Most of the XPS data have an unacceptably low signal-to-noise ratio, which significantly affects the reliability of the related analyses (for instance Figs. 1h

and 2d–e). Similarly, the Raman spectrum of Fig. S3 shows very weak Fe_3N signals. These analyses should be repeated.

Response: Thanks for your kind suggestion. We have repeated all these data as compared with previous counterpart.

(1) We have conducted XPS and Raman analysis again and the better XPS and Raman data have been obtained as shown in the Figures below. We would like to point out that low Fe_3N content, carbon coating and others will inevitably lead to the weak signals.

Figure R3. Repeated XPS data of Figure 1h, Figure 2d-e.

Figure R4. Repeated Raman data.

(2) In situ Raman is retested, as provide in the revised manuscript.

Figure 5f. In situ Raman spectra at a current density of 167.5 mA g⁻¹.

Updates to the revised manuscript: We have updated these Figures in the revised manuscript.

Comment 5: The Fe₃N ratio in the composite should be indicated in the manuscript.

Response: Thanks for your kind suggestion. TGA tests were employed to detect the loading of Fe₃N in the Fe₃N-NMCN composite. To clarify the transformation process of Fe₃N, TGA tests of the bare Fe₃N powder and Fe₃N-NMCN in O₂ atmosphere were conducted (Figure S8a-b). Moreover, the phase information of the final products after TGA tests was detected by XRD (Figure S8c-d). The bare Fe₃N powder was obtained through the heat treatment of FeCl₃·6H₂O at 300 °C for 1 h and then 800 °C for 2 h.

As displayed in Figure S8c-d, both products obtained after TGA tests are Fe₂O₃ (159.6

g mol⁻¹). When we assume that there is 1 mol Fe₃N (181.4 g mol⁻¹), there should be 1.5 mol Fe₂O₃ after the TGA test since the mole number of Fe is constant. During this process, the weight increment is 132.0wt.% (=159.6*1.5/181.4). This result is highly consistent with the TGA curve of the bare Fe₃N powder in Figure S8a, where the weight increase between room temperature and 800 °C is about 134wt.% and the main weight variation is in the range of 350 and 800 °C.

Therefore, it can be deduced that the weight loss (89wt.%) of Fe₃N-NMCN before 353 °C (Figure S8b) is exclusively originated from the consumption of carbon matrix in Fe₃N-NMCN and is not related to Fe₃N, while the weight increment (132wt.% = 14.5wt.%/11wt.%) between 353 and 800 °C is only from the oxidation of Fe₃N to Fe₂O₃. From above analyses, it has been determined that the weight percentage of Fe₃N in Fe₃N-NMCN is 11wt%.

Figure S8. Calculation of Fe₃N content in the Fe₃N-NMCN composite.

(a-b) TGA tests of the bare Fe₃N powder (a) and the Fe₃N-NMCN composite (b).

(c-d) XRD patterns of the final products after TGA tests from the bare Fe₃N powder (c) and the Fe₃N-NMCN composite (d).

Comment 6: This paper propose the use of the Fe₃N catalyst to achieve long-term cycling by mitigating the sulfur dissolution, although it does not provide sufficient experimental proof of the actual polysulfides (PSs) retention in the cathode during

cycling. In this regard, it might be worth remarking that various recent papers have demonstrated the full dissolution of the Li-PSs in the electrolyte, for instance by operando X-ray microscopy and diffraction (Energy Environ. Sci. 11, 202–210 (2018) and Adv. Energy Mater. 5, 1500165 (2015)), as well as in situ EPR and NMR (J. Electrochem. Soc. 162, A474–A478 (2015) and Nano Lett. 15, 3309–3316 (2015)), thereby partially addressing the conversion mechanism models proposed in the first stages of the alkali metal–S battery research.

Response: Thanks very much for recommending these valuable papers to help improve our manuscript. We have carefully studied these papers as Refs. 3-6, 72-74, and were inspired by these constructive viewpoints.

Indeed, it is widely accepted that polysulfides would partially or fully dissolved in the electrolyte and some sulphur radicals would be produced during the electrochemical reactions. In this case, the polysulfides would shuttle from the cathode to anode and resulting in the decay of cycling performance. In our work, two successive steps are essential to tackle this shuttling problem. Firstly, the Fe₃N nanodots, the doping N, the large surface and abundant pores of the Fe₃N-NMCN substrate provide powerful adsorption ability to these dissolved polysulfides, **which is confirmed by the adsorption test, UV-Vis spectra and DFT calculation.** Secondly, the trapped polysulfides are quickly catalyzed into short-chain products, which **is demonstrated by the symmetric CV and the DFT calculation.**

Additionally, these papers claimed that the resultant sulphur radicals would propel the proceeding of electrochemical reactions and different reaction pathways would happen during the discharge and following charge process. This is also supported by **our results obtained by in situ Raman (Figure 5f) and in situ XRD (Figure S36)** that S₈ changes into a series of polysulfides including Na₂S₈, Na₂S₆, Na₂S₄, Na₂S₂, Na₂S during the discharge process while only Na₂S₈ can be observed during the following charge process.

Updates to the revised manuscript: We have updated these discussions in the revised manuscript.

Figure 5f. In situ Raman spectra at a current density of 167.5 mA g^{-1} .

Figure S36. In situ XRD analysis of the $\text{S@Fe}_3\text{N-NMCN}$ electrode at a current density of 167.5 mA g^{-1} with an E/S ratio of 7.27 uL mg^{-1} (b) and corresponding

discharge/charge profile (a). (c-d) S 2p XPS spectra at the original state (a), discharged to 1.1 V (d) and discharged to 0.5 V (e).

References

- 1 Wang, Y. X. *et al.* Achieving high-performance room-temperature sodium-sulfur batteries with S@interconnected mesoporous carbon hollow nanospheres. *J. Am. Chem. Soc.* **138**, 16576-16579, doi:10.1021/jacs.6b08685 (2016).
- 2 Xia, G. *et al.* Carbon hollow nanobubbles on porous carbon nanofibers: an ideal host for high-performance sodium-sulfur batteries and hydrogen storage. *Energy Storage Mater.* **14**, 314-323, doi:10.1016/j.ensm.2018.05.008 (2018).
- 3 Wei, S. *et al.* A stable room-temperature sodium-sulfur battery. *Nat. Commun.* **7**, 11722, doi:10.1038/ncomms11722 (2016).
- 4 Wang, X., Zhang, Z., Qu, Y., Lai, Y. & Li, J. Nitrogen-doped graphene/sulfur composite as cathode material for high capacity lithium sulfur batteries. *J. Power Sources* **256**, 361-368, doi:10.1016/j.jpowsour.2014.01.093 (2014).
- 5 Yu, Q. *et al.* In situ formation of copper-based hosts embedded within 3D N-doped hierarchically porous carbon networks for ultralong cycle lithium-sulfur batteries. *Adv. Funct. Mater.* **28**, 1804520, doi:10.1002/adfm.201804520 (2018).
- 6 Liu, Y. *et al.* Nitrogen doping improves the immobilization and catalytic effects of Co₉S₈ in Li-S Batteries. *Adv. Funct. Mater.* **30**, 2002462, doi:10.1002/adfm.202002462 (2020).
- 7 Shan, Z. *et al.* Spontaneously rooting carbon nanotube incorporated N-doped carbon nanofibers as efficient sulfur host toward high performance lithium-sulfur batteries. *Appl. Surf. Sci.* **539**, 148209, doi:10.1016/j.apsusc.2020.148209 (2021).
- 8 Lai, W.-H. *et al.* General synthesis of single-atom catalysts for hydrogen evolution reactions and room-temperature Na-S batteries. *Angew. Chem., Int. Ed.* **59**, 22171–22178, doi:10.1002/anie.202009400.
- 9 Yan, Z. *et al.* Nickel sulfide nanocrystals on nitrogen-doped porous carbon nanotubes with high-efficiency electrocatalysis for room-temperature sodium-sulfur batteries. *Nat. Commun.* **10**, 4793, doi:10.1038/s41467-019-11600-3 (2019).

- 10 Zhang, B. W. *et al.* Long-life room-temperature sodium-sulfur batteries by virtue of transition-metal-nanocluster-sulfur interactions. *Angew. Chem., Int. Ed.* **57**, 1-6, doi:10.1002/anie.201811080 (2018).
- 11 Bao, W. *et al.* Boosting performance of Na-S batteries using sulfur-doped $\text{Ti}_3\text{C}_2\text{T}_x$ MXene nanosheets with a strong affinity to sodium polysulfides. *ACS nano* **13**, 11500-11509, doi:10.1021/acsnano.9b04977 (2019).
- 12 Zhang, B. W. *et al.* Atomic cobalt as an efficient electrocatalyst in sulfur cathodes for superior room-temperature sodium-sulfur batteries. *Nat. Commun.* **9**, 4082, doi:10.1038/s41467-018-06144-x (2018).
- 13 Guo, B. *et al.* Nickel hollow spheres concatenated by nitrogen-doped carbon fibers for enhancing electrochemical kinetics of sodium-sulfur batteries. *Adv. Sci.* **7**, 1902617, doi:10.1002/advs.201902617 (2020).
- 14 Zheng, J. *et al.* How to obtain reproducible results for lithium sulfur batteries? *J. Electrochem. Soc.* **160**, A2288-A2292, doi:10.1149/2.106311jes (2013).
- 15 Hagen, M., Fanz, P. & Tübke, J. Cell energy density and electrolyte/sulfur ratio in Li-S cells. *J. Power Sources* **264**, 30-34, doi:10.1016/j.jpowsour.2014.04.018 (2014).
- 16 Emerce, N. B. & Eroglu, D. Effect of electrolyte-to-sulfur ratio in the cell on the Li-S battery performance. *J. Electrochem. Soc.* **166**, A1490 (2019).
- 17 Sun, K. *et al.* Effect of electrolyte on high sulfur loading Li-S batteries. *J. electrochem. Soc.* **165**, A416 (2018).
- 18 Choi, J.-W. *et al.* Rechargeable lithium/sulfur battery with suitable mixed liquid electrolytes. *Electrochim. Acta* **52**, 2075-2082, doi:10.1016/j.electacta.2006.08.016 (2007).
- 19 Cui, Z., Zu, C., Zhou, W., Manthiram, A. & Goodenough, J. B. Mesoporous titanium nitride-enabled highly stable lithium-sulfur batteries. *Adv. Mater.* **28**, 6926-6931, doi:10.1002/adma.201601382 (2016).
- 20 Song, Y. *et al.* Synchronous immobilization and conversion of polysulfides on a VO_2 -VN binary host targeting high sulfur load Li-S batteries. *Energy Environ. Sci.* **11**, 2620-2630, doi:10.1039/c8ee01402g (2018).
- 21 Gao, W., Wang, Z., Peng, C., Kang, S. & Cui, L. Accelerating the redox kinetics by catalytic activation of “dead sulfur” in lithium-sulfur batteries. *J. Mater. Chem. A*, doi:10.1039/d1ta00772f (2021).
- 22 Xu, X. *et al.* A room-temperature sodium-sulfur battery with high capacity and stable cycling performance. *Nat. Commun.* **9**, 3870, doi:10.1038/s41467-018-06443-3 (2018).
- 23 Jin, F., Xiao, S., Lu, L. & Wang, Y. Efficient activation of high-loading sulfur

- by small CNTs confined inside a large CNT for high-capacity and high-rate lithium-sulfur batteries. *Nano Lett.* **16**, 440-447, doi:10.1021/acs.nanolett.5b04105 (2016).
- 24 Seh, Z. W., Sun, J., Sun, Y. & Cui, Y. A highly reversible room-temperature sodium metal anode. *ACS Cent. Sci.* **1**, 449-455, doi:10.1021/acscentsci.5b00328 (2015).
- 25 Di Lecce, D., Minnetti, L., Polidoro, D., Marangon, V. & Hassoun, J. Triglyme-based electrolyte for sodium-ion and sodium-sulfur batteries. *Ionics* **25**, 3129-3141, doi:10.1007/s11581-019-02878-w (2019).
- 26 Liang, H.-W. *et al.* Highly conductive and stretchable conductors fabricated from bacterial cellulose. *NPG Asia Mater.* **4**, e19-e19, doi:10.1038/am.2012.34 (2012).
- 27 Wan, Y. *et al.* Preparation and mineralization of three-dimensional carbon nanofibers from bacterial cellulose as potential scaffolds for bone tissue engineering. *Surf. Coat. Technol.* **205**, 2938-2946, doi:10.1016/j.surfcoat.2010.11.006 (2011).
- 28 Wang, B. *et al.* Pyrolyzed bacterial cellulose: a versatile support for lithium ion battery anode materials. *Small* **9**, 2399-2404, doi:10.1002/sml.201300692 (2013).
- 29 Qi, Y. *et al.* Slope-dominated carbon anode with high specific capacity and superior rate capability for high safety Na-ion batteries. *Angew. Chem., Int. Ed.* **58**, 4361-4365, doi:10.1002/anie.201900005 (2019).
- 30 Qi, Y. *et al.* Retarding graphitization of soft carbon precursor: From fusion-state to solid-state carbonization. *Energy Storage Mater.* **26**, 577-584, doi:10.1016/j.ensm.2019.11.031 (2020).
- 31 Li, Y., Yan, Y., Ming, H. & Zheng, J. One-step synthesis Fe₃N surface-modified Fe₃O₄ nanoparticles with excellent lithium storage ability. *Appl. Surf. Sci.* **305**, 683-688, doi:10.1016/j.apsusc.2014.03.169 (2014).
- 32 Zhang, F. *et al.* Metallic porous iron nitride and tantalum nitride single crystals with enhanced electrocatalysis performance. *Adv. Mater.* **31**, e1806552, doi:10.1002/adma.201806552 (2019).
- 33 Huang, H. *et al.* Fe₃N constrained inside C nanocages as an anode for Li-ion batteries through post-synthesis nitridation. *Nano Energy* **31**, 74-83, doi:10.1016/j.nanoen.2016.10.059 (2017).
- 34 Tang, X., Jia, R., Zhai, T. & Xia, H. Hierarchical Fe₃O₄@Fe₂O₃ core-shell nanorod arrays as high-performance anodes for asymmetric supercapacitors. *ACS Appl. Mater. Interfaces* **7**, 27518-27525, doi:10.1021/acsami.5b09766 (2015).

- 35 Jin, X. *et al.* Facile synthesis of monodispersed α -Fe₂O₃ cubes as a high-performance anode material for lithium-ion batteries. *Ionics*, doi:10.1007/s11581-021-04128-4 (2021).
- 36 Wu, Z. Y., Li, C., Liang, H. W., Chen, J. F. & Yu, S. H. Ultralight, flexible, and fire-resistant carbon nanofiber aerogels from bacterial cellulose. *Angew. Chem., Int. Ed.* **52**, 2925-2929, doi:10.1002/anie.201209676 (2013).

Reviewer #1 (Remarks to the Author):

The authors have given answers to the comments of the reviewers 1, 2, 3 (in a elaborate way) and accordingly the authors have modified the manuscript. The manuscript now is quite suitable for publication.

Reviewer #2 (Remarks to the Author):

Reviewer's comments:

The manuscript titled "Highly-efficient Fe₃N catalyst on N-doped hierarchical nanocarbons enables strong adsorption and fast dissociation of sodium polysulfides for advanced Na-S batteries", after a major revision is improved and I am pleased to see, that most of the open question raised during the first version are now answered.

I suggest this manuscript is accepted by Nature Communications after some minor revisions.

1. The reviewer suggest to write into the table the surface area and total pore volume, micropores volume, mesopores volume with less digits after the decimal separator. As example, the surface area into 410 m²g⁻¹ and pore volume with two digits (0.41 cm³g⁻¹)
2. The in situ XRD is a good techniques to identify crystalline products. Can the authors comment why Na₂S is not present into the XRD? One would expected that Na₂S is the final product and that is crystalline like Li₂S in Li-S batteries. How much was the archived capacity in the in situ XRD? Please put in the supplementary information the details for this measurements (in situ cell, window, electrochemistry and XRD parameters)

Reviewer #3 (Remarks to the Author):

The authors have significantly improved their manuscript by adding new measurements and further data in support to their conclusions. These additional results increase the expected relevance for the battery community of this work, which might become therefore suitable for publication in Nature Communications. On the other hand, the authors should further discuss some point that are not clear in the present version of the manuscript and address minor issues, as listed below.

- 1) The low volumetric energy density of practical cells is one of the current issues of metal-S batteries, and the relatively high thickness of the S-Fe₃N-NMCN electrode may adversely affect this cell parameter. The authors might further comment on this and compare the thickness of the Fe₃N-NMCN cathode with that of conventional electrodes cast on Al.
- 2) It is common belief that the polysulfides dissolved in the electrolyte solutions may improve the SEI on the lithium-metal anode in Li-S cells [Nat. Commun. 6, 7436 (2015)]. In view of this, the better surface morphology of the Na anode observed in the cells exhibiting lower polysulfide dissolution (Fig. S30) appears interesting and should be further commented.
- 3) The authors provided additional proof of polysulfide retention by Fe₃N, nitrogen doping, and cathode porosity and described the advantages of polysulfide-adsorption approaches. However, possible alternative strategies to mitigate the detrimental effects of Na₂S_x dissolution, such as the anode protection, should be mentioned in the manuscript.
- 4) Various figures as well as discussion shown in the rebuttal letter are not reported in the manuscript, that is, the XPS analyses (Figs. 1h, 2d, and 2e) as well as the discussion on the electrolyte formulation (Reviewer #1, comment #4). The authors should update the manuscript accordingly. Furthermore,

various responses to the Reviewers' comments have been often repeated, and the letter is unnecessarily long.

Reviewer #1

The authors have given answers to the comments of the reviewers 1, 2, 3 (in a elaborate way) and accordingly the authors have modified the manuscript. The manuscript now is quite suitable for publication.

Response: Thank you very much for your highly constructive comments on the manuscript.

Reviewer #2:

The manuscript titled “Highly-efficient Fe₃N catalyst on N-doped hierarchical nanocarbons enables strong adsorption and fast dissociation of sodium polysulfides for advanced Na-S batteries”, after a major revision is improved and I am pleased to see, that most of the open questions raised during the first version are now answered. I suggest this manuscript is accepted by Nature Communications after some minor revisions.

Response: We thank the reviewer for the positive and detailed comments on our revised manuscript and we appreciate your further inputs.

Comment 1: The reviewer suggest to write into the table the surface area and total pore volume, micropores volume, mesopores volume with less digits after the decimal separator. As example, the surface area into 410 m² g⁻¹ and pore volume with two digits (0.41 cm³ g⁻¹)

Response: Thanks very much for your valuable suggestion. We have reduced digits after the decimal separators. Please refer to the highlighted part in the revised manuscript and supporting information, Pages 8, 16, 21-22, Page 9 in Supporting information.

Comment 2: The in situ XRD is a good technique to identify crystalline products. Can the authors comment why Na₂S is not present into the XRD? One would expect that Na₂S is the final product and that is crystalline like Li₂S in Li-S batteries. How much was the archived capacity in the in situ XRD? Please put in the supplementary information the details for this measurements (in situ cell, window, electrochemistry and XRD parameters)

Response: We appreciate your constructive comment to help improve our manuscript. Our *in situ* XRD is conducted under a low E/S ratio of 7.27 uL mg⁻¹ and thus the

corresponding capacity of the *in situ* cell is only 731.5 mA h g⁻¹, a little lower than that reported in the manuscript (1165.9 mA h g⁻¹ under an E/S ratio of 18.2 uL mg⁻¹). There are three reasons for the fact that Na₂S is not presented in the *in situ* XRD pattern. First, only a few Na₂S are formed under this low E/S ratio and thus the signal of Na₂S is weak. Second, the generated Na₂S can be covered by other intermediates. Third, a conventional XRD instrument with limited resolution and power was employed to conduct the *in situ* experiment. Therefore, the diffraction peaks of Na₂S can hardly be observed by the *in situ* XRD.

The *in situ* cell was purchased from Bruker (Beijing) Technology Co., LTD. A Be foil was used to seal the cell. The *in situ* XRD was conducted at a current density of 167.5 mA g⁻¹ and an electrochemical window of 2.8 - 0.5 V on X-Ray Diffraction (XRD, Bruker, Advance D8A A25). Each XRD spectrum was collected within 12 min.

Updates to the revised manuscript: We have added these details/discussions in the revised manuscript on **Page 36** and supporting information on **Pages 27-28**, highlighted in yellow.

Reviewer #3:

The authors have significantly improved their manuscript by adding new measurements and further data in support to their conclusions. These additional results increase the expected relevance for the battery community of this work, which might become therefore suitable for publication in Nature Communications. On the other hand, the authors should further discuss some point that are not clear in the present version of the manuscript and address minor issues, as listed below.

Response: Thank you very much for your affirmation. We appreciate your further comments.

Comment 1: The low volumetric energy density of practical cells is one of the current issues of metal-S batteries, and the relatively high thickness of the S-Fe₃N-NMCN electrode may adversely affect this cell parameter. The authors might further comment on this and compare the thickness of the Fe₃N-NMCN cathode with that of conventional electrodes cast on Al.

Response: Thanks very much for your valuable suggestion to improve our manuscript. Indeed, the relatively high thickness of S@Fe₃N-NMCN may adversely lead to low volumetric energy density of practical metal-S batteries, which is a mutual problem of free-standing electrodes. We think this problem can be mitigated by controlling the electrode preparation process, e.g., using an effective calendaring-infiltration strategy to achieve a more compact electrode, meanwhile retaining high reversible specific capacity.

As shown by FESEM images in Figure S12 and Figure S39, the electrode still contains many pores (sulfur active materials are infiltrated into micro/mesopores of carbon fibers, whereas some interstices between carbon layers still exist). To overcome this issue, we have successfully developed an effective calendaring-infiltration strategy to achieve a more compact electrode. The Fe₃N-NMCN composite was pressed and carbon disulfide solution of sulfur was added onto the composite. The mass ratio of sulfur to the composite was kept at 10:1. During the following drying/heating process, the composite loaded with sulfur was also pressed.

As displayed by FESEM images in Supplementary Figure 10, the calendaring-infiltration method leads to a close packing of carbon fibers and therefore a thinner electrode with a thickness of 115 μm . The thickness of such electrode is comparable with those electrodes casted on Al with similar S mass loading (Carbon, 2014, 75, 161-168; Carbon, 2017, 111, 493-501). In this case, the obtained compact electrode still exhibits a high reversible specific capacity of 959 mA h g^{-1} . The above result demonstrates that a compact and thin electrode with excellent performance can be obtained through controlling the preparation process.

Updates to the revised manuscript: We have added these results in the revised manuscript on Page 33 and supporting information on Page 30, highlighted in yellow.

Supplementary Figure 10 FESEM images (a-b) and voltage-capacity curve (c) of the compact S@Fe₃N-NMCN electrode obtained through a calendaring-infiltration strategy.

Comment 2: It is common belief that the polysulfides dissolved in the electrolyte solutions may improve the SEI on the lithium-metal anode in Li-S cells [Nat. Commun. 6, 7436 (2015)]. In view of this, the better surface morphology of the Na anode observed in the cells exhibiting lower polysulfide dissolution (Fig. S30) appears interesting and should be further commented.

Response: We appreciate your nice suggestion to further improve our manuscript. We have carefully read the suggested paper and cited as Ref. 74. In the suggested paper (Nat. Commun. 6, 7436 (2015)), the authors stated that a synergistic effect of lithium polysulfide and lithium nitrate in ether-based electrolyte can lead to a stable and uniform solid electrolyte interphase (SEI) layer on Li metal. However, it was also

recognized by the authors that polysulfides alone cannot prevent the dendrite growth and minimize the electrolyte decomposition. On the contrary, polysulfides alone have bad effect on metal anodes.

In our work, only polysulfide is presented in the electrolyte and no synergistic effect can be generated to realize a uniform SEI on Na metal anode in a polysulfides-rich environment. In view of this, a better surface morphology of the Na anode (for S@Fe₃N-NMCN) compared with S@BC-Ar is resulted from the less shuttling of polysulfides to the Na anode. The excellent bonding ability to polysulfides achieved by Fe₃N, nitrogen doping and cathode porosity can restrain the dissolved polysulfides in the cathode area. Thus, little polysulfides can enter into the anode area and deteriorate the Na anode.

Updates to the revised manuscript: We have commented this point on Page 28, highlighted in yellow.

Comment 3: The authors provided additional proof of polysulfide retention by Fe₃N, nitrogen doping, and cathode porosity and described the advantages of polysulfide-adsorption approaches. However, possible alternative strategies to mitigate the detrimental effects of Na₂S_x dissolution, such as the anode protection, should be mentioned in the manuscript.

Response: Thanks very much for your suggestion. Besides, other alternative strategies such as the anode protection (Nat. Commun. 6, 7436 (2015); Energy Storage Mater. 3 77-84) and the electrolyte/separator modification (Nano Lett. 20, 5391-5399; Small Methods. 4, 2000082) can be combined to further mitigate the detrimental effects of dissolved Na₂S_x, enabling a better performance and making the as-proposed electrode more practical.

Updates to the revised manuscript: We have added this discussion in the revised manuscript on Page 31, highlighted in yellow.

Comment 4: Various figures as well as discussion shown in the rebuttal letter are not reported in the manuscript, that is, the XPS analyses (Figs. 1h, 2d, and 2e) as well as the discussion on the electrolyte formulation (Reviewer #1, comment #4). The authors should update the manuscript accordingly. Furthermore, various responses to the Reviewers' comments have been often repeated, and the letter is unnecessarily long.

Response: Thanks very much for your helpful comments. We have checked the manuscript seriously and updated accordingly. Figs. 1h, 2d, and 2e are updated by the repeated data. In addition, the discussion on the electrolyte formulation (Reviewer #1, comment #4) has been also addressed in the revised manuscript, as highlighted on Pages 22-23.